# Analysis of Lightning-Induced Currents in Supply Cable Shields and Their Impact on LLS Sensor Site Errors

2 3

1

#### Hannes Kohlmann<sup>1,2</sup>, Wolfgang Schulz<sup>1</sup>, Farhad Rachidi<sup>2</sup>, Naiara Duarte<sup>3</sup>, Dmitry Kuklin<sup>4</sup> 4

- <sup>1</sup>OVE-ALDIS, Vienna, Austria 5
- 6 <sup>2</sup>EPFL, EMC Laboratory, 1015 Lausanne, Switzerland
- <sup>3</sup>CEFET-MG Department of Electrical Engineering, Belo Horizonte, Brazil 7
- 8 <sup>4</sup>NERC KSC RAS Apatity, Murmansk Region, Russia

9 10

Correspondence to: Hannes Kohlmann (h.kohlmann@ove.at)

11

20

34

35

36

#### 12 Abstract

Lightning location system (LLS) sensors, which detect and locate atmospheric discharges, are typically powered by cables buried up to one meter underground. Within the LLS community, it is well known that these cables can create spurious magnetic fields, which can in turn adversely impact the sensor measurements and the resulting data. This issue arises from currents induced in the cable shield by the lightning electromagnetic fields that penetrate the ground. The magnetic field generated by these currents lead to "site errors," causing inaccuracies in estimating the angle of incidence and the peak current of lightning strokes. Although these sensor-specific errors can be partially corrected, a better understanding of the coupling mechanism between the lightning electromagnetic field and the cable could help in minimizing the site errors. This study presents an analysis of the lightning electromagnetic field interaction with cables and examines the influence of various ground and cable properties on this interaction. This work represents a first step toward understanding the physical mechanism leading to LLS sensor site errors. Considering simplified scenarios involving a single insulated or bare conductor, this work provides practical insights that LLS operators can use to estimate worst-case site errors for a provisioned sensor site. Additionally, we show that some site errors observed in operational sensors can be successfully reproduced with good agreement using the proposed approach.

#### 1 Introduction

- Lightning location systems (LLSs) include a network of sensors whose purpose it is to detect and 30 geolocate lightning discharges. Their main functional principle is based on electric and/or magnetic field 31 sensors detecting an incident electromagnetic (EM) field generated by a lightning discharge. To estimate
- the strike point location, two techniques can be utilized, either individually or combined (see, for
- example, Chapter 13 in (Cooray et al., 2022)):
  - 1) Time-of-arrival (ToA): The strike position is estimated using multilateration based on the time difference of arrival at different sensors. The arrival times are determined using precise, GPSsynchronized time stamps.

2) Magnetic direction finding (MDF): The strike position is determined by intersecting the estimated directions of the incident field at multiple sensors. This technique is often combined with the ToA technique to achieve optimum positioning results.

The MDF technique relies on H-field measurements through crossed coils yielding voltages that represent the amplitude of two (x, y) or three (x, y, z) components of the incident H-field. Recently, it has been shown that estimation errors of the angle of the incident field can be related to the propagation terrain by reflection and diffraction phenomena caused by hills and mountains, see for example (Kohlmann et al., 2021).

a) Field-to-cable coupling mechanism

42

52

56

60

b) Shield currents and scattered H-field

Fig. 1: Mechanism of lightning EM field coupling to a buried LLS sensor power supply cable

The present study, in contrast, addresses specific issues related to the MDF technique, which are wellknown to the LLS community since the beginning of the application of MDF, namely "angle site errors" and "amplitude site errors" (see e.g., (Schulz, 1997), (Schulz and Diendorfer, 2002)). They are related to spurious additive magnetic fields, caused by induced currents on the buried power supply cable or nearby conductive objects. These spurious magnetic fields superimpose on the main incident field, leading to inaccuracies. This type of interference affecting the measurements originates from the very incident field that the LLS sensor is designed to detect. Specifically, the lightning EM field exhibits a horizontal E-field component in the direction of propagation (typically referred to as E<sub>r</sub>, but projected onto the x-axis in the present study, thus at the concerned sections also referred to as E<sub>x</sub>), resulting from the finite conductivity of the nearby ground. As this field penetrates the ground, it interacts with any metallic structure, such as buried pipes, bare or insulated conductors, cable shields, etc., inducing electric currents. The attenuation of the E-field while penetrating the ground impacts the amplitude of the coupled currents, thus the burial depth of the cable also plays a role, albeit not dominant as the present study will show. The induced currents generate a scattered magnetic field (referred to as  $\vec{H}_{err}$  in this study), which, when in close proximity of the H-field sensor, superimpose on the incident lightning magnetic field. An illustration of this electromagnetic environment near an LLS sensor is presented in Fig. 1.

The orientation of the scattered magnetic field  $\vec{H}_{err}$  depends on the relative position of the underground cable or metallic structure, leading to the distortion in the ratio of the x- and y-components of the detected H-field at the time the sensor samples the signal. Thus, the sensor may estimate an incorrect angle of incidence, which is a critical parameter in lightning location systems that rely on the MDF technique. In the LLS community, this error is referred to as the "angle site error" or "angle error". Apart from that, the addition of a magnetic field component, affects the measured magnitude of the lightning H-field. This is referred to as "amplitude site error" or "signal error". Since the measured magnitude is used to estimate the peak current of the lightning return stroke, the peak current estimate is also affected, typically leading to an overestimation. This occurs because the amplitude error is positive in most cases, as will be shown in this study. The angle and amplitude errors are tightly related to each other as the results presented in Section 3.3 of the present study show (see also (Schulz and Diendorfer, 2002)). As indicated in Fig. 1b, the induced current in the cable shield can have forward and backward propagating waves, the latter being dependent on the termination impedance (i.e., current reflection coefficient). Ideally, a disconnected shield from the ground would yield the smallest currents near the H-field sensor, resulting in minimal angle and amplitude site errors. In real life, however, the cable shield is not always disconnected from the ground. It is the engineer's task to control and minimize these effects as much as possible, by carefully connecting the sensor to the power supply and thoroughly evaluating the local conditions at the sensor site. This includes the connection of the power supply cable (and potentially a separate communications cable) to the internal circuitry of the sensor's power cabinet, taking into account the protective earth (PE) wire(s), cable shield(s), and sensor grounding (through earth electrodes) and other structures related to the installation of an LLS sensor. Therefore, a thorough and in-depth understanding of the sensor's electromagnetic environment and the underlying physical factors causing LLS sensor site errors is of paramount importance.

64

68

81

87

Since site errors at each sensor can be empirically evaluated during a thunderstorm season through reference to the optimum positioning results of the whole LLS network with high location accuracy (which is in the order of 100 m, see (Schulz et al., 2016)), the systematic correction of the site errors is a relatively straightforward task. Consequently, even in the presence of large errors, the correction methods enable angle estimates that contribute meaningfully to location algorithms. However, the qualitative and quantitative characteristics of these errors are still only roughly understood.

Fig. 2: Theoretical angle site errors (Azimuth difference, top) and amplitude site errors (Error [%], bottom), merely based on simplified geometrical considerations. The shown curves are evaluated assuming different shield current amplitudes, with higher currents corresponding to higher site errors. (Graphic reused with permission from (Schulz and Diendorfer, 2002)).

It is typically observed in practice that site errors related to induced currents on buried cables exhibit a double-cycle sinusoidal-type of curve, as depicted in Fig. 2. This behavior is expected because during a full azimuthal rotation of the incident field, the induced currents necessarily become zero for particular angles, such as when the field impinges perpendicularly on the power supply cable. For the rest of the angles, sine and cosine functions determine the amplitude of the electromagnetic field impinging on the cable, giving rise to the observed sinusoidal behavior. While in reality, angle site errors of varying levels have been observed, ranging from exceptionally low values below  $\pm$  1°, to more typical levels of  $\pm$  3° to  $\pm$ 5°, and even exceeding  $\pm$  10° in extreme cases. Many of these errors exhibit asymmetric azimuthal behavior due to the complex electromagnetic environment near the sensor and the surrounding topographic terrain (see (Kohlmann et al., 2021)).

The primary objective of the present work is to demonstrate, for the first time, the emergence of site errors caused by buried power supply cables, through a semi-analytical approach involving (a) determining the radial lightning EM field at the ground surface and below the ground, (b) computing the cable currents induced by these fields, and, (c) calculating the resulting scattered magnetic fields caused by these induced currents. In other words, the study aims to show that field-to-cable coupling, aside from the terrain-related site errors, is one of the main physical mechanisms contributing to the site errors observed in the LLS sensors utilizing the magnetic fields to locate lightning and estimate the peak current. We will show that the scattered magnetic fields due to the induced currents on the cable shield can, under specific scenarios of electrical connection of the entering power supply cable, replicate the

level of angle and amplitude site errors observed in actual sites. Furthermore, this work aims to identify the most important parameters impacting the magnitude of these site errors, such as the ground conductivity, the power supply cable length, the vertical distance of the cable to the sensor and the cable grounding method.

In this study, a single insulated solid wire, acting as a proxy for a shielded conductor, is considered for the investigation of LLS sensor angle and amplitude site errors. The influence of bare wires is also investigated for comparison. The methods used are comparable to those described in (Aguet et al., 1980), (Bridges, 1995) and (Bridges, 1992), but are adapted to consider incident EM plane waves (with grazing angles of incidence), which are associated with remote lightning strikes.

The rest of the paper is organized as follows. Section 2 presents the methodology followed to obtain the results of this study. It elaborates on the steps to compute remote lightning EM fields (above and below ground), which are used in the following step as the input to the field-to-cable current coupling model. Then, the model for the field-to-cable current coupling and the approach used to evaluate the magnetic fields generated by the induced cable currents is described, as well as their subsequent impact on the resulting site errors. Section 3 presents the results of the individual computation steps. They include key graphs illustrating the expected level of vertical and horizontal E-fields, induced currents, scattered magnetic fields and resulting site errors. Compound graphs are also provided to help readers estimate the maximum expected site errors based on parameters such as supply cable length, ground conductivity and shield termination impedance. Section 4 discusses the practical relevance of the presented results, evaluates the agreement between theoretical predictions and experimental observations, and highlights some relevant real-world insights from the experience of LLS operators. The conclusion summarizes the work and provides an outlook on future work.

### 2 Methodology

- In this Section, we present the procedure for the computation of LLS sensor angle and amplitude site errors, which is carried out in three steps: (a) calculation of the lightning electric fields along the cable, (b) computation of the induced current in the cable shield, and (c) computation of the resulting scattered magnetic field, and the evaluation of the site errors.
- 2.1 Return stroke modelling, lightning EM field propagation and ground penetration
- In order to investigate the induced currents for incident fields typical for lightning discharges, the remote fields associated with the lightning return stroke (RS) have to be obtained in a first step. The geometry of the problem is illustrated in Fig. 3. The objective is to compute the horizontal electric field along the buried cable (z = -d), which will serve as source term in the field-to-cable coupling equations (see next subsection). The lightning return stroke is assumed to be a straight vertical antenna located at 100 km from the cable (typical distance covered by LLS sensors). The average ground conductivity along the propagation path is assumed to be  $\sigma_p$ , while the local ground conductivity at the sensor site is  $\sigma_{loc}$ . This

latter will determine the amplitude and waveshape of the horizontal E<sub>x</sub>-field driving the cable shield current.

Fig. 3: Lightning EM field propagation towards the sensor site at a distance of 100 km.

To represent the lightning return stroke channel, the Modified Transmission Line model with Exponential Decay (MTLE, (Nucci et al., 1988) and (Rachidi and Nucci, 1990)) was used. The parameters of the model were set to  $\lambda = 2$  km (exponential decay of the RS current with height),  $v_{RS} = 1.5 \cdot 10^8$  m/s (RS wavefront speed). The channel-base current is represented by the sum of two Heidler's functions, described by the following formula:  $I(0,t) = \frac{I_1}{\xi_1} \frac{\left(\frac{t}{\tau_{11}}\right)^{n_1}}{\left(\frac{t}{\tau_{11}}\right)^{n_1}+1} e^{-\frac{t}{\tau_{12}}} + \frac{I_2}{\xi_2} \frac{\left(\frac{t}{\tau_{21}}\right)^{n_2}}{\left(\frac{t}{\tau_{21}}\right)^{n_2}+1} e^{-\frac{t}{\tau_{22}}}$ ,  $\xi_i = \exp\left(-\frac{\tau_{i1}}{\tau_{i2}}\left[n_i\frac{\tau_{i2}}{\tau_{i1}}\right]^{1/n_i}\right)$ . The parameters were chosen to form a channel-base current with characteristics of a typical subsequent RS:  $I_1 = 10.7$  kA,  $\tau_{11} = 0.25$  µs,  $\tau_{12} = 2.5$  µs,  $I_2 = 6.5$  kA,  $\tau_{21} = 2$  µs,  $\tau_{22} = 230$  µs and  $\tau_{12} = 2$  (see (Rachidi et al., 2001)). The corresponding subsequent RS-type current waveform, with its short rise time of less than 1 µs, is depicted in Fig. 4.

Fig. 4: Return stroke current waveform representing a typical subsequent RS.

The vertical electric field and the horizontal magnetic field generated by the return stroke are first computed assuming a propagation over a perfectly electric conducting (PEC) ground. The computation is performed according to (Thottappillil et al., 1997), where the contributions of current dipoles along the channel are summed up to obtain the fields at an observation point located on the ground surface.

To account for the attenuation and dispersion that affects the lightning EM fields while propagating above a lossy ground, the fields computed assuming a PEC ground can be corrected by applying specific filters as described, e.g., in (Norton, 1937), (Wait, 1953), (Shoory et al., 2010) (see also (Wait, 1970) for a thorough compendium on wave propagation effects of electromagnetic fields along stratified media). Cross-validations using cylindrically symmetric 2D-FDTD (finite-difference time-domain) simulations have shown the best agreement using the Wait's attenuation function for a stratified ground, as described in details in (Shoory et al., 2010). While the use of Wait's function allows to straightforwardly consider a horizontally stratified ground, the case of a homogeneous (single-layer) ground was considered assuming a very thick upper layer (e.g., 10 km) to account accurately for the attenuation for arbitrary distances without spurious reflection phenomena from the lower layer boundary (for poor ground conductivities). The according equations (see (Shoory et al., 2010) for details) can be readily implemented by typical numerical computational libraries. The Wait's expression for the attenuation function is given by

$$F_{str}(p_{str}) = 1 - j\sqrt{\pi p_{str}} e^{-p_{str}} \operatorname{erfc}(j\sqrt{p_{str}}),$$

$$\text{in which}$$

$$p_{str} = -0.5 \, \gamma_0 d \, \Delta_{str}^2$$

$$\gamma_0 = j\omega \sqrt{\mu_0 \varepsilon_0}$$

$$\Delta_{str} = \sqrt{\frac{\varepsilon_0}{\mu_0}} K_1 \frac{K_2 + K_1 \tanh(u_1 h_1)}{K_1 + K_2 \tanh(u_1 h_1)}$$

$$K_i = u_i / (\sigma_i + j\omega \varepsilon_0 \varepsilon_{ri}), \ i \in 1,2$$

$$u_i = \sqrt{\gamma_i^2 - \gamma_0^2}, \ i \in 1,2$$

$$\gamma_i = \sqrt{j\omega \mu_0 (\sigma_i + j\omega \varepsilon_0 \varepsilon_{ri})}, \qquad i \in 1,2$$

$$\gamma_i = \sqrt{j\omega \mu_0 (\sigma_i + j\omega \varepsilon_0 \varepsilon_{ri})}, \qquad i \in 1,2$$

where d is the propagation distance,  $h_1$  is the thickness of the upper layer of the stratified ground. The sub-index i denote the parameters of the respective layer,  $i \in 1,2$ , and sub-index 0 denotes parameters of the free space. The ground conductivity of the top layer, is referred to as  $\sigma_p$  (with index 'p' denoting 'propagation') throughout the paper and impacts the rise time of the propagating EM field.

The next step is to determine the radial E-field  $(E_r)$  in the direction of propagation, at the ground level and below the ground surface. To achieve this, the procedure described in (Rubinstein, 1996) is implemented, in which the so-called "wave-tilt" formula (Rubinstein, 1996) is used to obtain the radial E-field at the ground surface from the horizontal H-field as determined in the previous step, through the surface impedance of the air-ground interface:

$$E_r(z=0) = -H_{\varphi}(z=0) \sqrt{\frac{\mu_0}{(\varepsilon_g + \frac{\sigma_{loc}}{i\omega})}}$$
 (2)

with  $\varepsilon_g$  being the ground permittivity,  $\sigma_{loc}$  the local ground conductivity,  $\mu_0$  the magnetic permeability of free space and  $\omega$  the angular frequency. Finally, the electric field at a depth z below the ground level is found using Weyl's formulation, which reads

$$E_r(z) = E_r(z=0) \cdot e^{j\omega \sqrt{\mu_0 \left(\varepsilon_g + \frac{\sigma_{loc}}{j\omega}\right)} z}$$
 (2 < 0)

These three formulations (1), (2) and (3) lead to very accurate results, as confirmed by comparisons with full wave numerical simulations using the FDTD method (see also Section 3.1). Finally, the radial Efield  $E_r$  is projected onto the direction of the cable by multiplying with the cosine of the angle between the cable's direction and  $E_r$ . These techniques provide us with the means to accurately compute the impinging horizontal electric fields on the cable, which serve as inputs for the field-to-cable coupling equations. Since  $\sigma_{loc}$  has a significant influence on the horizontal E-field, the coupling mechanism and, ultimately, the resulting LLS sensor site errors, values on the order of the expected (local) ground conductivities should be assumed when simulating a particular site. Although strong variations in local ground conductivities are generally expected even within small volumes near the cable (see for example (Rizki Ramdhani et al., 2020) or (Loke, 2001)), regional ranges of estimated ground electrical conductivity values are available in the World Atlas of Ground Conductivities (ITU Radiocommunication Assembly, 1999).

## 2.2 Field-to-cable coupling

An essential component of the angle and amplitude site error investigations is the field-to-cable coupling model, which uses frequency-domain solutions based on Green's functions. These functions incorporate the coupling equations, while horizontal electric fields act as distributed sources along the cable length, as detailed in (Aguet et al., 1980) and (Tesche et al., 1997). Various approaches for coupling models have been explored, such as the treatment of bare and insulated wires for infinitely long lines in (Bridges, 1995), shielded cables with multiple layers and terminations in (Aguet et al., 1980), and finite-difference time-domain methods for buried conductors and cable shields subjected to lightning strikes in (Petrache et al., 2005). Further discussions on generated electric and magnetic fields in buried cables can be found in (Bridges, 1992). Bridges derived exact solutions for the induced current on an infinite bare or insulated cable buried in soil due to a transient plane wave (Bridges, 1995) and demonstrated that the transmission line theory provides accurate results for a wide range of cases. An experimental validation for the accuracy of the transmission line theory for field-to-cable coupling computations is presented in (Paolone et al., 2005).

The relation describing the induced currents at a point x along a buried cable of length L, using Green's function  $G_I(x,x_s)$ , reads (see for example (Petrache et al., 2005)):

$$I(x) = \int_0^L G_I(x, x_s) E_x^e(x_s, z = -d) dx_s$$
 (4)

where  $E_x^e$  is obtained from Eq. (3) presented earlier and the Green's function reads:

223224

226227

$$G_{I}(x,x_{s}) = \begin{cases} \frac{e^{-\gamma L}}{2Z_{c}(1-\rho_{1}\rho_{2}e^{-2\gamma L}} \left[e^{-\gamma(x_{s}-L)} - \rho_{2}e^{\gamma(x_{s}-L)}\right] \cdot \left[e^{\gamma x} - \rho_{1}e^{-\gamma x}\right] \text{ for } x < x_{s} \\ \frac{e^{-\gamma L}}{2Z_{c}(1-\rho_{1}\rho_{2}e^{-2\gamma L}} \left[e^{-\gamma(x-L)} - \rho_{2}e^{\gamma(x-L)}\right] \cdot \left[e^{\gamma x_{s}} - \rho_{1}e^{-\gamma x_{s}}\right] \text{ for } x > x_{s} \end{cases}$$
(5)

The equations involve the complex propagation constant  $\gamma = \sqrt{Z'Y'}$ , the cable characteristic impedance  $Z_c = \sqrt{Z'/Y'}$ , the line length L and the voltage reflection coefficients  $\rho_i = (Z_i - Z_c)/(Z_i + Z_c)$ , with  $i \in 1,2$ , at the line ends, whereby  $Z_1$  and  $Z_2$  are the source and load impedances of the cable respectively. Due to the different expressions for  $x < x_s$  and  $x > x_s$  in Eq. (5), the integral in Eq. (4) needs to be partially integrated:  $I(x) = \int_0^{L-x} ... dx_s + \int_{L-x}^L ... dx_s$ , resulting in an analytical expression that can be straightforwardly calculated.

The longitudinal impedance Z' and transverse admittance Y' involve the calculation of the perunit-length ground impedance  $Z'_g$  and ground admittance  $Y'_g$  (see in Fig. 5). In the present study, Theethayi's ground impedance formulation was used (see Equation (9) in (Theethayi et al., 2007)). Detailed discussions on ground impedance models are available in (Petrache et al., 2005) and (Theethayi et al., 2007), while advanced formulations that account for the soil parameter frequency dependence are found in (Visacro and Alipio, 2012) and (Duarte et al., 2021). For the convenience of the reader, the expressions for the distributed parameters (Fig. 5), as described in (Theethayi et al., 2007), are reproduced as follows.

Fig. 5: Equivalent circuit based on the Transmission Line model of an infinitesimal element of the cable (left: insulated, right: bare) in presence of an external electromagnetic field (tangential E-field, E<sub>x</sub>)

$$Z'_g = \frac{j\omega\mu_0}{2\pi} \left\{ \ln\left(\frac{1+\gamma_g R_{ab}}{\gamma_g R_{ab}}\right) + \left[\frac{2e^{-2d|\gamma_g|}}{4+\gamma_g^2 R_{ab}^2}\right] \right\}$$
 (6)

$$Y'_g \approx \frac{\gamma_g^2}{Z'_g} \tag{7}$$

in which  $\gamma_g = \sqrt{j\omega\mu_0(\sigma_g+j\omega\varepsilon_g)}$  is the complex propagation constant in the ground.

$$L' = \frac{\mu_0}{2\pi} \cdot \ln\left(\frac{\rho_b}{\rho_a}\right) \text{ (for an insulated wire)}$$
 (8)

$$C' = 2\pi \varepsilon_{ins} / \ln(\frac{\rho_b}{\rho_a})$$
 (for an insulated wire) (9)

where,

- for insulated wires, ρ<sub>a</sub> is the inner wire radius, ρ<sub>b</sub> is the outer radius (including the dielectric jacket of permittivity ε<sub>ins</sub>), R<sub>ab</sub> = ρ<sub>b</sub>;
- for bare wires,  $R_{ab} = \rho_a$ .
- For insulated wires, the total per-unit cable series impedance is  $Z' = j\omega L' + Z'_{g}$ , and the total per-unit
- shunt admittance is  $Y' = \frac{CY'_g}{j\omega C + Y'_g}$ , while for bare wires, the total per-unit series impedance is
- $Z' = Z'_g$ , and the total per-unit shunt admittance is  $Y' = Y'_g$  (see Fig. 5).
  - All the equations were implemented in the frequency domain and applied under the assumption that both the input signal and the resulting outputs are real, causal signals. The input signal (horizontal magnetic field  $H_{\varphi}$  above PEC) spectrum was obtained using Fast Fourier Transform (FFT), with frequencies considered up to half the Nyquist frequency. To reconstruct real, causal signals, the upper half of the frequency spectrum (from half the Nyquist frequency to the Nyquist frequency) was completed by appending the complex conjugate of the computed results from the lower half of the spectrum. The final time-domain signal was then obtained by performing an inverse FFT on the completed frequency-domain data.

# 2.3 Scattered magnetic field generated by the induced current on the cable shield and the resulting angle/amplitude site-errors

a) Three-dimensional view of the electromagnetic environment in the vicinity of the H-field sensor, positioned at the cable head at (x,y,z)=(L,0,h). The supply cable is oriented in the x-direction. The incident field is indicated together with its propagation velocity vector  $\vec{v}$  (with  $|\vec{v}|=c_0$ ) and its field components  $\vec{H}_{\phi}$ ,  $\vec{E}_z$  and  $\vec{E}_r$ . The x-directed shield current, induced by the incident field, is indicated by the red arrows along the cable and denoted as  $I_{\rm sh,x}$ .

b) Top-down view (xy-plane) of Fig. 6a centered at the H-field sensor at (x,y,z) = (L,0,h). The direction of the incident field is marked by the dotted arrow. The horizontal magnetic component of the incident EM field,

 $\vec{H}_{\varphi}$  (black), the scattered field,  $\vec{H}_{err}$  (red), and the sampled field  $\vec{H}_{sampled} = \vec{H}_{\varphi} + \vec{H}_{err}$ , (blue) are also shown. The angle error is denoted as  $\alpha_{err}$  (red) and the amplitude error results from the difference between the magnitudes of the vectors  $\vec{H}_{sampled}$  and  $\vec{H}_{\varphi}$ .

Fig. 6: Illustrations of LLS angle and amplitude site errors caused by the induced shield currents in the power supply cable.

As shown in (Bridges, 1992), the calculated currents along the line can be straight-forwardly used to compute the nearby magnetic fields, as they are not strongly impacted by the air-earth interface. Consequently, the scattered magnetic field is computed using Biot-Savart's law, integrating the contributions of the current elements, obtained in the previous step (Section 2.2), along the nearest 50 m to the magnetic field sensor for each time instant. To solve the spatial integral by summing the contributions of the small current elements, it is important to have an accurate spatial current function. This can be readily achieved using a quadratic interpolation function. Moreover, due to the  $1/r^2$  distance dependency, contributions beyond 50 m are assumed to be negligible. Fig. 6 illustrates the mechanism of site errors. The scattered field is denoted as  $\vec{H}_{err}(I_{sh,x})$ , exhibiting y- and z-components in the given geometrical arrangement. The cable is oriented in the x-direction, as the field depends on the induced shield, respectively conductor current  $I_{sh,x}$ , which is aligned with the cable's direction.

This field adds a spurious term to the incident field  $\vec{H}_{\phi}$  (horizontal, thus purely in the xy-plane) resulting in a sampled field  $\vec{H}_{\rm sampled}$  that exhibits an altered angle and magnitude in comparison to the true incident field  $\vec{H}_{\phi}$ . As a consequence of Ampere's law, the error magnetic field vector  $\vec{H}_{err}$  is azimuthal around the power supply cable, which is assumed to be straight. The vertical component of the head of

the supply cable below the sensor, and thus the corresponding H-field, is not considered, as it is aligned axially with the sensor and is assumed to have a negligible impact. For more complex shapes of the power supply cable paths, including corners and bends), the scattered (error) field vector  $\vec{H}_{err}$  may have arbitrary orientations. Further, it must be noted that Fig. 6 must be understood as a snapshot at a specific sampling instant, where all vector lengths and angles are time-dependent according to the incident EM wave and the induced currents. A typical LLS sensor samples the field when the vector  $\vec{H}_{sampled}$  reaches its maximum value, referred to as maximum sampled magnitude in the present study. At this instant, the difference between the true incident field angle  $\phi$  and the sampled angle,  $\phi_{sampled}$ , computed using the arc tangent of the output voltage ratio of the x- and y-component of the crossed loop antenna, is defined as the angle error  $\alpha_{err}$ . The amplitude error (sometimes also called "signal error") is denoted as  $s_{err}$  and defined as  $s_{err} = \frac{|\vec{H}_{sampled}|}{|\vec{H}_{\phi}|}$ .

#### 3 Simulation results

## 3.1 Lightning EM fields and ground penetration

This section presents the simulation results of lightning incident electric fields following the procedure described in Section 2.1, considering a channel-base current waveform that exhibits characteristics that are typical of subsequent return strokes (in particular, characterized by a short rise time), as depicted in Fig. 4. All results are obtained for a distance to the lightning discharge of 100 km. Due to the linearity and time-invariance of the equations utilized in this paper, the amplitude of the channel-base current was kept constant throughout all computations. Variations of the E-fields used as input for the coupling analyses were solely the result of the assumed ground parameters along the propagation path (see Fig. 7. The main results of this paper, namely the angle and amplitude site errors, are independent of the selected channel-base current amplitude; that is, they are unaffected by any scaling of the waveform.

Fig. 7: Distant vertical electric field (100 km) at the ground level as a function of the ground electric conductivity (PEC,  $\sigma_p = 10 \cdot 10^{-3}$  S/m,  $1 \cdot 10^{-3}$  S/m and  $0.1 \cdot 10^{-3}$  S/m).

The vertical electric field above the PEC is shown in Fig. 7 (blue curve). Note that the vertical E-field  $(E_z)$  and the horizontal H-field  $(\vec{H}_{\varphi})$ , which is sensed by an MDF sensor) above a PEC ground are related through  $H_{\varphi} = c_0/\mu_0 \cdot E_z$ . Simulation results using Wait's formalism, accounting for the attenuation of

the fields due to the propagation over a lossy ground, are also shown in Fig. 7 for different ground conductivities. As can be seen, the higher the conductivity, the lower the attenuation and dispersion. Lower values for the ground conductivity lead to more attenuated and dispersed fields with longer rise times (about 2  $\mu$ s, 4  $\mu$ s and 10  $\mu$ s for the orange, green and red curve, respectively). It is worth noting that the frequency-dependent attenuation function (1) is also a function of distance: the farther the field propagates, the greater the attenuation and dispersion. Thus, for closer lightning strikes, the fields retain more of the high-frequency content and exhibit shorter rise times, e.g., at 50 km compared to those depicted in Fig. 7 for 100 km.

 The radial E-fields at the ground level and below ground are obtained after applying Eq. (2), and Eq. (3) to the horizontal magnetic field above lossy ground. The results are shown in Fig. 8.

The impact of the burial depth for higher ground conductivities, namely  $\sigma_{loc} = 10 \cdot 10^{-3}$  S/m and  $\sigma_{loc} = 50 \cdot 10^{-3}$  S/m, for an incident field according to Fig. 7 ( $\sigma_p = 10 \cdot 10^{-3}$  S/m, orange curve), is shown in Fig. 8a and Fig. 8b. As can be seen, a significant reduction in the amplitude is already observed within the first 5 meters below the ground level.

Fig. 8: Variation of the horizontal electric field as a function of the burial depth for higher values of the ground conductivity  $\sigma_{loc}$  with  $\sigma_p = 10 \cdot 10^{-3}$  S/m. Compare to Fig. 9.

Since these horizontal E-fields serve as the input for the cable current coupling model, ensuring their validity is crucial for achieving accurate results in subsequent computational steps. Thus, the fields were validated through cross-comparison with results obtained from a cylindrical-symmetry 2D-FDTD solver ((Oskooi et al., 2010), (Anon, 2024)). The results for both approaches are found to be in excellent agreement. A validation example is shown in Fig. 9. The depicted scenario represents very low ground conductivities  $(0.1 \cdot 10^{-3} \text{ S/m})$  and  $0.01 \cdot 10^{-3} \text{ S/m}$  and  $0.01 \cdot 10^{-3} \text{ S/m}$  and large depths (d = -50 m) below ground. The difference in the amplitudes between the fields at ground level and those below ground is very small, indicating that attenuation in low-conductivity ground is negligible for typical burial depths of power supply cables, which range from a few tens of centimeters to about 1 m.

Fig. 9: Validation of the proposed approach with respect to FDTD simulations. Shown are the E-fields at the ground level and 50 m below ground for very low ground conductivities  $\sigma_{loc} = 10^{-4}$  S/m and  $\sigma_{loc} = 10^{-5}$  S/m, with  $\sigma_p = 10^{-4}$  S/m. The time-axis represents the absolute time of arrival of the EM field at a distance of 100 km (approximately 333  $\mu$ s).

Fig. 10 illustrates the general dependencies of the horizontal electric fields, both at the ground level and below ground, on the propagation path ground conductivity. Fig. 10a shows the ratio of the peak value of the horizontal electric field at ground level,  $E_{x,peak}(d = 0 \text{ m})$  to that of the vertical E-field,  $E_{z,peak}$ , at the ground level. This ratio can be as high as 30% for very low conductivity ( $\sigma_{loc} = 0.01 \cdot 10^{-3}$  S/m) and drops to 2.5% for high ground conductivity ( $\sigma_{loc} = 50 \cdot 10^{-3} \text{ S/m}$ ). Due to the frequency dependence of the physical mechanisms governing the local field configuration, these ratios depend on the frequency content of the incident field. To account for attenuation and dispersion along the 100-km propagation different lossy grounds with ground conductivity values ranging from  $\sigma_p = 100 \cdot 10^{\text{--}3}$  S/m to  $\sigma_p = 0.1 \cdot 10^{\text{--}3}$  S/m have been investigated.

a) Ratio of the horizontal  $E_x$ -field peak to the vertical  $E_z$ -field peak at the ground surface level, as a function of the local ground conductivity  $\sigma_{loc}$ , in %. Plotted for different propagation ground conductivities  $\sigma_p$ 

316317

b) Ratio of the peak value of the Ex-field at a depth of d meters below ground to that at the ground surface level. The solid lines represent a propagation conductivity  $\sigma_p = 10 \cdot 10^{-3}$  S/m while the dashed lines correspond to  $\sigma_p = 1 \cdot 10^{-3}$  S/m.

Fig. 10: Field peak ratios for various local ground conductivities  $\sigma_{loc}$  and the impact of the propagation ground conductivity  $\sigma_p$ 

As previously shown in Fig. 7, propagation over a highly conducting ground (ideally PEC) preserves the high-frequency content of the propagating fields. This results in incident fields exhibiting fast

transients and corresponding short risetimes. In contrast, propagation over less conductive ground attenuates the high-frequency content and causes dispersion, leading to incident fields with slower transients and longer risetimes. Examination of Fig. 10a now reveals two key aspects. (1) fields with shorter rise times (fast transients) produce larger  $E_x$ -field peaks (as evidenced by the bold blue curve with the thin red curve at a given local ground conductivity  $\sigma_{loc}$ ) and (2) low local ground conductivity produces large  $E_x$ -field peaks, whereas highly conductive local ground reduces the  $E_x$ -field peak significantly that eventually reaches zero for infinite ground conductivity  $\sigma_{loc}$  (PEC ground). A realistic scenario for a lightning EM field involves propagation over lossy ground with conductivity values  $\sigma_p$  between  $0.1 \cdot 10^{-3}$  S/m and  $10 \cdot 10^{-3}$  S/m over 100 km, resulting in incident fields similar to those shown in Fig. 7.

The penetration of the horizontal E-field,  $E_x(d=0\,\text{m})$  into various depths below ground level is shown in Fig. 10b, illustrating the ratio of the  $E_x$ -field peaks at the surface and below ground. The figure clearly shows that low ground conductivities allow for deep penetration, with minimal attenuation of the field peak over depth. Conversely, for high conductivities such as  $\sigma_{loc} = 50 \cdot 10^{-3} \text{S/m}$ , the attenuation with depth below ground is more significant. Notably, e.g., at d=1m below ground, attenuation ranges between 13% (dashed, thickest blue line) to 20% (solid, thickest blue line) – a critical observation discussed further in Section 4. The discrepancy between the solid and dashed lines again stems from the fact that waveforms with a higher frequency spectrum (i.e., fast transients) are better preserved during propagation along a medium with high conductivity  $\sigma_p$  during propagation, are more significantly attenuated through the ground at the sensor site, irrespective of the local ground conductivity  $\sigma_{loc}$ .

#### 3.2 Field-to-cable current coupling

Hereafter in this section, the field-to-cable coupling computations described in Section 2.2 are performed assuming a propagation ground conductivity  $\sigma_p = 10 \cdot 10^{-3}$  S/m and a local ground conductivity  $\sigma_{loc} = 10 \cdot 10^{-3}$  S/m, unless stated otherwise. The conductor radius is  $\rho_a = 10$  mm, while the cable jacket is 5 mm thick, resulting in an outer radius of  $\rho_b = 15$  mm. The cable jacket has a relative permittivity of  $\epsilon_{r,d} = 3$ . The conductor can be regarded as a cable shield with an equivalent outer radius typical of power supply cables. The impact of a slightly higher electrical resistance due to a thin screen, as opposed to a solid conductor, is negligible in the coupling analyses that follow.

Fig. 11: Shield currents of an insulated cable of 450 m length at various locations x responding to a distant (100 km) lightning-incident field, as shown in Fig. 7 for  $\sigma_p = 10 \cdot 10^{-3}$  S/m,  $\sigma_{loc} = 10 \cdot 10^{-3}$  S/m. Burial depth d = 1 m. The incident angle is  $\varphi = 0^{\circ}$  relative to the cable (aligned in x-direction).

Two examples considering different termination impedance scenarios are considered. The results for the induced currents are shown in Fig. 11. Fig. 11a presents the results for a cable terminated at its ends with an impedance of  $Z_1 = Z_2 = 10~\Omega$ , a value typically achievable at sites with  $\sigma_{loc} = 10 \cdot 10^{-3}$  S/m. The peak current at the line end (at x = 450 m, thickest lines in Fig. 11) reaches a value of about 100 mA. Fig. 11b shows the simulation results assuming that the line end is disconnected (shield not connected to ground modeled through a large impedance value of  $Z_2 = 10~M\Omega$ ). In this configuration, the current at x = 450 m is naturally zero. However, within the first 50 m away from the line end (see x = 400 m), the current gradually increases, reaching a peak value of about 60% of the overall peak current, which occurs near the middle of the line. Since a completely ungrounded shield (with no connection to ground on either side of the cable) is an uncommon installation practice for shielded cable, this scenario is not considered. Such a setup would also result in zero current at the start of the line (x = 0 m).

Fig. 12: Currents at the line end with  $Z_1 = Z_2 = 10 \ \Omega$  (same parameters as in Fig. 11a), considering different line lengths L and a local site conductivity of  $\sigma_{loc} = 10 \cdot 10^{-3} \ \text{S/m}$ . Burial depth d = 1 m. The incident angle is  $\phi = 0^{\circ}$  relative to the cable (aligned in x-direction)

The effect of the line length on the induced current is shown in Fig. 12. As the line length decreases, a corresponding reduction in the induced shield current peak is observed. A significant

increase of the induced shield current peak current would be observed when the local conductivity is lower. This observation is aligned with the findings of Section 3.1, where a reduction in the local conductivity results in an increased horizontal electric field and decreased attenuation during ground penetration. However, in scenarios with reduced conductivities, the smallest achievable line termination impedance also increases, compensating the current increase and reducing the currents near the line ends. The largest shield current amplitudes would in turn be closer to the midsections of the cable.

The induced currents also depend on the conductor radius, generally exhibiting higher amplitudes for larger radii. However, these differences are in the order of  $\pm 10\%$  when the conductor radius is doubled. Therefore, the influence of conductor radius on the results is not further analyzed in this study.

## 3.3 Scattered magnetic field and angle/amplitude site errors

In this section, the waveshape of the scattered magnetic field  $\vec{H}_{err}$  (se Fig. 6) is examined more in detail, and the related angle and amplitude site errors that would, at least theoretically, be observed by the MDF sensor used in LLSs is explored. The impact of several parameters, such as line length, ground conductivity, burial depth and the vertical separation between the MDF antennae and the supply cables is investigated.

Fig. 13: Dependence of the magnetic fields on the local ground conductivity. All magnetic field components are shown for an incident EM field (compare Fig. 6, whereby this graph has been adapted to depict all fields with positive polarity). The incident angle is  $\phi=30^\circ$  relative to the cable (aligned in x-direction). The resulting angle errors  $\alpha_{err}$  and amplitude errors  $s_{err}$  are presented on the top of each figure. The insulated cable length is L=200 m and buried 1 m below ground. Conductor radius:  $\rho_a=10$  mm, outer radius  $\rho_b=15$  mm, cable jacket permittivity  $\epsilon_{r,d}=3$ . Sensor position (line end) at z=+2 m. 100 km distance to the lightning strike.  $\sigma_p$  was set to  $1\cdot 10^{-3}$  S/m. The DC termination impedances  $Z_{1,2}$  correspond to a vertical grounding rod of 10 m length and 3 cm thickness for the considered ground conductivity  $\sigma_{loc}$ .

As explained in Section 2.2, once the spatiotemporal behavior of the conductor currents is determined, the scattered magnetic field can be computed by applying Biot-Savart's law. Given the short distances of the relevant current elements impacting the sensed magnetic field (less than 50 m), time retardation can be disregarded. Fig. 13 presents the results for four local ground conductivity ( $\sigma_{loc}$ ) scenarios as described in the figure caption. To get reasonable results, the termination impedances  $Z_1$  and  $Z_2$  were assumed to be conductivity- and frequency-dependent, considering a 10-m long, 3-cm thickness vertical grounding rod (see for example (Grcev, 2009)). Their DC-values are given in the subfigures of Fig. 13. Otherwise, the shield currents would reach unrealistically high amplitudes due to a grounding impedance value which would be unattainable at a site with, for example, a very low conductivity. The graphs show the main lightning field to be detected,  $\vec{H}_{\phi}$  (black dotted line), incident with an angle of  $\phi = 30^{\circ}$  relative to the cable (aligned with the x-axis). They also show the x- and y-components (orange and green) of  $\vec{H}_{\phi}$ , the scattered field  $\vec{H}_{err}$ , which is a y-component that adds to Hy, due to a current  $I_{sh}$  oriented along the x-axis, and finally the resulting total field  $\vec{H}_{sampled}$ , which the

sensor samples at the instant of its maximum magnitude. The scattered field  $\vec{H}_{err}$  is responsible for a distortion of the H-field vector of the true incident lighting EM field, resulting in angle and amplitude errors,  $\alpha_{err}$  and  $s_{err}$ , respectively. Note that the estimated time of arrival is not significantly affected by the addition of the  $\vec{H}_{err}$  field, as it is determined as close as possible to the onset of the waveform's rising edge. Thus, the LLS location results obtained using the ToA technique remain unaffected by the phenomena illustrated in Fig. 13.

397398

405406

408

410

415

The differences in site errors shown in Fig. 13 a-d is attributed to the characteristics of the scattered fields  $(\vec{H}_{err})$  impacting the site errors at the sampling instant. For high conductivity values (e.g.,  $\sigma_{loc} = 10 \cdot 10^{-3}$  and  $1 \cdot 10^{-3}$  S/m, see Fig. 13a and Fig. 13b), the maximum of the scattered field  $\vec{H}_{err}$ nearly coincides with the maximum of  $\vec{H}_{sampled}$ . However, for very low ground conductivity (0.1·10<sup>-3</sup> S/m and smaller, see Fig. 13c Fig. 13d), the induced current wave on the cable shield experiences minimal attenuation as it propagates along the shield. This leads to pronounced reflections and resonances along long lines. As a consequence, oscillations arise in the induced currents, producing fast ringing effects in the scattered field  $\vec{H}_{err}$ . These oscillations exhibit a frequency that depends on the line length and can arbitrarily impact the sampling instant - and thus the site errors potentially causing sampling to occur when  $\vec{H}_{err}$  is zero or even of opposite polarity. It must be noted that the sharper the impinging transients, the more pronounced the oscillations of the induced current response. While incident fields with very high-frequency content (i.e., short rise times), combined with very low local ground conductivity  $\sigma_{loc}$  and long cables, may occur in reality, such scenarios are rare. Nevertheless, this possibility should not be overlooked, because, as explained in Section 3.1, lightning discharges occurring close to the sensor also contain high frequency content, and thus short measured rise times can be expected.

Fig. 14: Induced currents and scattered magnetic fields: insulated vs. bare conductors. L = 100 m,  $\rho_a$  = 10 mm,  $\rho_b$  = 15 mm, cable jacket permittivity  $\epsilon_{r,d}$  = 3. Sensor position (line end) at z = +2 m.  $\sigma_p$  = 1·10<sup>-3</sup> S/m,  $\sigma_{loc}$  = 10·10<sup>-3</sup> S/m,  $\epsilon_{r,g}$  = 10 and  $Z_2$  = 10  $\Omega$ . The incident angle is  $\varphi$  = 30°.

If an insulated power supply cable were hypothetically replaced by a bare conductor of the same length, parameters, and termination conditions, simulations indicate that the induced conductor currents near the line end would be moderately reduced. The associated scattered field  $\vec{H}_{err}$  would, in turn, be reduced as well, resulting in reductions in site errors. Specifically, the angle site error decreases by 33%, and the amplitude site error by 37%, as shown in the example presented in Fig. 14a and Fig. 14b and shows the result for a cable of length L=100 m. The results of currents and H-fields were compared to results obtained from fully integrated 3D-FDTD simulations, using the open source FDTD-solver Elecode (see (Kuklin, 2022)), which supports modeling insulated conductors. The comparisons showed very good agreement.

Next, after thoroughly addressing the physical quantities, including the electric fields below ground, coupled currents and scattered magnetic fields, that contribute to the LLS sensor site errors, it remains to finally investigate the angle and amplitude site errors over a full  $360^{\circ}$  azimuth rotation. Hereby, key parameters impacting the results will be highlighted. The H-field sensor is assumed to be located at a height of z = 2 m above ground.

We begin by examining the impact of the burial depth of the power supply cable on the site errors. The simulation results are presented in Fig. 15 and cover two distinct scenarios:

• Scenario 1: As the burial depth increases, the distance between the cable to the sensor head also increases, reflecting the most realistic scenario. In this case, the site error reduction is influenced by a combined effect of increasing distance between the cable to the H-field sensor and the field attenuation by the ground (solid lines in Fig. 15).

• Scenario 2: The cable is buried at different depths, but the relative distance between the cable and the H-field sensor is kept constant at 2 meters. This scenario isolates the effect of ground attenuation from the distance effect, highlighting their distinct contribution. The impact of ground attenuation alone is shown in dashed lines in Fig. 15.

Fig. 15: Impact of the burial depth on LLS sensor site errors for  $\sigma_p = 10 \cdot 10^{-3}$  S/m and  $\sigma_{loc} = 10 \cdot 10^{-3}$  S/m, L = 100 m. Solid lines: Combined effect of field attenuation and increased distance to the sensor. Dashed line: Impact of field attenuation below ground, while the distance of the cable to the sensor is kept constant (z = 2 m). (a) angle site errors  $\alpha_{err}$ , (b) amplitude site errors  $s_{err}$ 

The results presented in Fig. 15 were obtained for a local ground conductivity  $\sigma_{loc} = 10 \cdot 10^{-3}$  S/m. They reveal a significant finding: The site errors are very strongly impacted by the (vertical) distance of the cable to the H-field sensor, as indicated by the solid-line curves. In contrast, the dashed-line curves, representing the scenario with a fixed 2-m distance, exhibit only a minor reduction in site errors with increasing burial depth. Specifically, at a burial depth of 1.5 m in Scenario 2, the angle site error  $\alpha_{err}$  is reduced by only 8.5%. However, in Scenario 1, where the cable-to-sensor distance increases with burial depth, the reduction reaches 46%. This finding is consistent with results presented in Fig. 10b which suggests the same effect based on the attenuation caused by the ground penetration of the  $E_x$ -field for the assumed parameters. The amplitude site errors  $s_{err}$  exhibit a similar trend, decreasing by comparable amounts.

Next, the impact of a significantly higher local ground conductivity  $\sigma_{loc}$  is investigated. As shown previously in Fig. 10b, higher conductivity increases the attenuation of the illuminating Ex-field as it penetrates to ground. Additionally, Fig. 10a demonstrated that higher  $\sigma_{loc}$  leads to smaller site errors due to the reduced horizontal Ex-field illuminating the cable shield. To account for this effect, a new baseline angle site error was calculated for a cable placed at ground level (d = 0 m) and a sensor located 2 m above, assuming a value for the local ground conductivity of  $\sigma_{loc} = 50 \cdot 10^{-3}$  S/m. The angle site error in this case drops to 3.86°, compared to 7.5° for  $\sigma_{loc} = 10 \cdot 10^{-3}$  S/m at an azimuth of 130°, for example. Using this new baseline angle site error, the impact of ground attenuation for a buried cable is reevaluated. For Scenario 2 (only the effect of ground attenuation), the angle site error is reduced by 20%

at a burial depth of d=1.5 m, compared to just 8.5% for the lower conductivity case  $\sigma_{loc}=10\cdot 10^{-3}$  S/m. In Scenario 1 (which includes both ground attenuation and increased distance to the sensor), the reduction reaches 54%, compared to 46% for  $\sigma_{loc}=10\cdot 10^{-3}$  S/m.

Thus, while the attenuation-caused reduction is greater for higher  $\sigma_{loc}$  (20% vs. 8.5%), the dominant factor contributing to the total site error reduction in Scenario 1 remains the increased vertical distance between the sensor and the cable. It is important to note that these findings are independent of the significant overall decrease in site error of almost 50% (for  $\sigma_{loc} = 50 \cdot 10^{-3}$  S/m) in contrast to  $\sigma_{loc} = 10 \cdot 10^{-3}$  S/m) that results directly from the reduced Ex-field strength at high local ground conductivity.

At this point, one further investigation naturally presents itself: examining the impact of the increasing sensor's vertical distance to the cable, as the sensor may be mounted on top of a high mast. This installation type has been employed at some LLS sensor sites, where it has been associated with minimal site errors. It is expected that the observed behavior will approximate an inverse relationship to the distance 1/r, with r being the vertical distance from the cable to the sensor. This expectation aligns with Ampere's law, according to which the magnetic field of an infinitely long cable is  $H = I/(2\pi r)$ . This relationship is confirmed in Fig. 16, which shows the angle site error decreasing from  $\pm$  6.7° at a vertical distance of 2.5 m from the cable to  $\pm$  1.8°. Although slightly less pronounced, this reduction closely follows the expected 1/r relationship. One important comment must be added: further induction phenomena are expected due to the prevailing vertical E-field ( $E_z$ ) impinging on the high mast and the cable. This could lead to potential additional induced currents in the grounding of the mast, which could impact the shield currents of the supply cable and, consequently, the behavior illustrated in Fig. 16. The results match well with real-world experience of this type of installation, which was previously employed by the Austrian LLS operator ALDIS (Austrian Lightning Detection and Information System).

Fig. 16: Dependence of the angle site error on the vertical distance of the H-field sensor to the supply cable. Cable burial depth d=0.5 m,  $\sigma_{loc}=10\cdot 10^{-3}$  S/m. The maximum angle errors should be interpreted as  $\pm$  the values shown on the y-axis, according to the two-cycle error characteristic.

The final evaluation aims to provide a comprehensive overview of the influence of the local ground conductivity, length and termination impedance - factors previously illustrated by means of time-domain graphs of coupled currents and magnetic fields - on the site errors, summarized in a single figure. All

c) Maximum amplitude site errors a<sub>err</sub> (insulated conductor wire)

 d) Maximum amplitude site errors serr (bare wire)

Fig. 17: Influence of local ground conductivity  $\sigma_{loc}$ , cable length L and (conductivity- and frequency-dependent) termination impedances  $Z_1$  and  $Z_2$  on site errors.  $\sigma_p = 1 \cdot 10^{-3}$  S/m, burial depth d = 0.5 m. Sensor position at z = 2 m. The order of the color legend agrees with the maximum site errors according to the line lengths: largest site errors for L = 600 m on top of the legend, monotonically decreasing to the smallest site errors for L = 25 m at the bottom of the legend.

(a) maximum angle site errors  $\alpha_{err}$  (insulated conductor), (b) maximum angle site errors  $\alpha_{err}$  (bare wire), (c) maximum amplitude site errors  $\alpha_{err}$  (insulated conductor wire), (d) maximum amplitude site errors  $\alpha_{err}$  (bare wire)

simulations were performed considering an insulated wire with an outer cable radius of  $\rho_b = 7.5$  mm and core radius of  $\rho_a = 5.8$  mm, a jacket dielectric relative permittivity of  $\epsilon_{r,d} = 3$ , buried at a depth of 50 cm. Two different sets of ground parameters are considered to examine the impact of different (short vs. long) rise times of the incident lightning fields: (i)  $\epsilon_{r,g} = 10$ ,  $\sigma_p = 1 \cdot 10^{-3}$  S/m, and, (ii)  $\epsilon_{r,g} = 10$ ,  $\sigma_p = 0.1 \cdot 10^{-3}$  S/m. The distance to the lightning strike was assumed to be 100 km. Both line ends are assumed to be grounded with a vertical grounding rod (see (Grcev, 2009)), resulting in termination impedances which are frequency- and conductivity-dependent. This approach provides a more realistic representation compared to a constant grounding impedance, which may not appropriately represent the prevailing local grounding conditions. Having both ends grounded, where the largest currents can flow, represents the worst-case scenario for LLS sensor site errors. This analysis

deliberately focuses on this worst-case scenario with the aim of understanding the primary interrelation

c) Maximum amplitude site errors  $\alpha_{err}$  (insulated conductor wire)

497

500

d) Maximum amplitude site errors serr (bare wire)

Fig. 18: Same as Fig. 17, but with  $\sigma_p = 0.1 \cdot 10^{-3}$  S/m, further reducing the frequency content of the incident lightning EM field and thereby increasing the rise time (see Fig. 7). The order of the color legend agrees with maximum site errors according to the line lengths: largest site errors for L = 600 m on top of the legend, monotonically decreasing to the smallest site errors for L = 25 m at the bottom of the legend.

(a) maximum angle site errors  $\alpha_{err}$  (insulated conductor), (b) maximum angle site errors  $s_{err}$  (bare wire), (c) maximum amplitude site errors  $\alpha_{err}$  (insulated conductor wire), (d) maximum amplitude site errors  $s_{err}$  (bare wire)

isolated and hypothetical scenario should be taken with care and regarded solely as a rough estimate for the maximum expected site errors. The discussion section is dedicated to further considerations and to the elaboration on more special or real-world case studies

Fig. 17 a-d ( $\sigma_p = 1 \cdot 10^{-3}$  S/m) and Fig. 18 a-d ( $\sigma_p = 0.1 \cdot 10^{-3}$  S/m) depict the maximum observable angle site errors  $\alpha_{err}$  (occurring at an incident angle of about  $\phi = 45^{\circ}$ , refer to Fig. 6 for comparison) and amplitude site errors  $s_{err}$  (occurring at  $\phi = 0^{\circ}$ ) for different incident fields – one with faster rise times (Fig.17) and one with slower rise times (Fig-18). Fig. 17 and Fig. 18 a & c show  $\alpha_{err}$  and  $s_{err}$  for insulated cables, b & d for bare wires. The graphic also shows the direct current impedance value  $Z_{DC}$ , which the

line termination impedances  $Z_1$  and  $Z_2$  assume for a vertical grounding rod of 10 m length, on the right ordinate. The frequency dependence was taken into account as well, although the impact is negligible for the frequency range of the induced currents.

The most important observations are summarized in the following bullet points, based on the use of "fast fields" for the case with  $\sigma_p = 1 \cdot 10^{-3}$  S/m (i.e., Fig. 17), with rise times of about 4 µs (Fig. 7, green curve), due to the higher frequency components present in the incident field, and "slow fields" for  $\sigma_p = 0.1 \cdot 10^{-3}$  S/m (i.e., Fig. 18) with rise times of about 10 µs (Fig. 7, red curve), reflecting the low frequency content of the incident field:

- Decreasing the supply cable length decreases site errors but causes a slight shift in the maximum site errors towards higher conductivity values. This trend is consistent for incident fields of different rise times.
- The effects visible for fast fields and low local ground conductivities  $\sigma_{loc}$  in conjunction with long lines are attributed to resonance effects (see Fig. 13d). In this case, small four-cycle site errors and a slight underestimation of the lightning peak current can be observed. However,
  - these phenomena are not observed for incident fields with longer rise times (see Fig. 18 corresponding with rise times of about 10 µs).
- For long power supply cables, slow fields result in greater site errors than fast fields. This might
  be attributed to the extended time available for propagation effects of the induced current wave
  on the cable shield to impact the total current at the line's endpoint, leading to higher current
  values and, consequently, larger site errors. In contrast, for short lines (L < 100 m), the
  simulation results exhibit no dependence on rise time.</li>
- The almost identical values of the maximum site errors for different line lengths in the case of fast fields (Fig. 17) becomes noticeable for line lengths exceeding approximately 200 m. This is because for high local ground conductivity, dissipation prevents significant (unattenuated) propagation of the induced current wave on the cable shield until the time that the sampling is performed by the sensor. Henceforth, remote current induction effects are not detected by the sensor. For lower values of  $\sigma_{loc}$ , shield current wave propagation effects are present, leading to a divergence of the curves below  $\sigma_{loc} = 1 \cdot 10^{-3}$  S/m. For slow fields (Fig. 18), the saturation is observed for longer cables (exceeding 450 m). This is because, by the time of the sampling of  $H_{sample}$  at about 10  $\mu$ s, the wave can, in contrast to fast fields, propagate farther and build up currents close enough to the sensor.
- The decrease in amplitude and angle errors on the right-hand side of the bell-shape site error curves, where the grounding impedance Z<sub>DC</sub> (resulting from high σ<sub>loc</sub>) is very low, is constrained by the diminishing E<sub>x</sub>-field components caused by the high local ground conductivity (see Fig. 10a). To the left of the peak site errors, the site errors decrease due to the high grounding

impedance  $Z_{DC}$ . However, as the local ground conductivity, which would reduce current dissipation along the line, decreases, the shield current wave propagation effects become more pronounced. Consequently, long lines are more susceptible to higher site errors, and even polarity reversal effects for  $\alpha_{err}$  and amplitude attenuation (negative  $s_{err}$ ) may occur for low ground conductivity. Within the considered range of conductivities  $\sigma_{loc}$ , these effects are observed only for fast fields with sharper transients (Fig. 17), but not for slow fields (Fig. 18).

• Bare wires, being in contact with the ground, dissipate propagating currents much more efficiently. This can explain, why the site error shows no significant dependence on the rise time of the field (compare Fig. 17 b & d, with Fig. 18 b & d).

The graphs in Fig. 17 a & c and Fig. 18 a & c have practical application. For a given site provisioned for sensor installation, the LLS operator can easily estimate the expected maximum site errors. These graphs represent the worst-case scenario, where the cable shield of a supply cable (insulated conductor scenario) is grounded at both sensor ends. For a given cable length L and a vertical cable-to-sensor distance of 2.5 m, the maximum angle error  $\alpha_{err}$  or amplitude error  $s_{err}$  can determined based on the local ground conductivity  $\sigma_{loc}$ , and the sensor grounding impedance  $Z_{DC}$ . If the sensor grounding impedance is lower than the  $Z_{DC}$  value (blue ordinate in Fig. 17 and Fig. 18) for the given conductivity, the maximum site errors will exceed those shown in the graphs (due to higher currents at lower impedance). Conversely, if the grounding impedance is higher, the site errors will be smaller.

#### 4 Discussion

This section serves as the ground to discuss the phenomenon of LLS sensor site errors, both in general and in relation to how they align with the practical experience of LLS operators.

While supply cable-related LLS sensor site errors exhibit a two-cycle periodicity, they are not fully symmetric, as suggested in (Schulz et al., 1998) and shown in Fig. 2. Specifically, they are not two-cycle sinusoidal. This asymmetry is more pronounced for long insulated supply cables, in which, when the angles of incidence align with the cable's orientation, the E-field interacts with a larger segment of the cable. It results in an induced current wave that propagates as a travelling wave along the cable, which, upon reaching close proximity to the sensor, significantly affects the site error. Conversely, if the EM wave approaches from the opposite direction, reaching the sensor first, currents are gradually induced, and the current elements along the cable take effect later in time, resulting in a lesser impact on the site errors. Consequently, both angle and amplitude site errors,  $\alpha_{err}$  and  $s_{err}$ , are slightly lower for angles of incidence  $90^{\circ} 

Fig. 19: (a) shows the mean site error (red solid line) of real measured site errors in a scatter plot of individual pixels that indicate the number of detected lightning EM fields from low (blue) to high (yellow/orange), and is compared to simulation results in (b), based on the methodology presented in this paper. Parameters:  $\sigma_p = 0.2 \cdot 10^{-3} \, \text{S/m}$  to obtain incidents fields of about 8-9 µs rise time (according to the median measured rise time at the sensor site).  $\sigma_{loc} = 10 \cdot 10^{-3} \, \text{S/m}$ ,  $\epsilon_{r,g} = 10$ . A 600-m long insulated power supply cable, oriented at  $\phi = 290^{\circ}$ , is assumed to be buried 20 cm below ground in flat, swampy open terrain. The cable shield ends are both connected to ground.

An important effect is observed, when the insulated wire is replaced with a bare conductor. This change leads to an effective reduction in the angle and amplitude site errors, as illustrated in Fig. 14. In (Theethayi and Thottappillil, 2007), the interaction between a horizontal grounding electrode and parallel power supply and communication cables is discussed. This interaction may help explain why the measured shield currents in (Schulz et al., 1998) were significantly lower than those predicted by the present study. In (Schulz et al., 1998), the measured shield current magnitude was about 28 mA, while the incident (vertical)  $E_z$ -field was approximately 6 V/m – twice the magnitude considered here. This implies that for a field strength of 3 V/m (as shown in Fig. 7), the shield current would be approximately 14 mA. This value is substantially lower than the simulated results presented in Fig. 12, which assume a line length of 200 m and a local conductivity of  $\sigma_{loc} = 10 \cdot 10^{-3}$  S/m. Even with an unrealistically high value of  $\sigma_{loc} = 50 \cdot 10^{-3}$  S/m, the computed shield current would still be much higher

than the measured value. The findings in (Theethayi and Thottappillil, 2007) suggest that a horizontal ground electrode of about 10 m length and aligned with the power supply and communication cable, may have favorably influenced the results by reducing site errors observed in (Schulz et al., 1998). Future studies should consider the impact of a follow-on bare wire, such as horizontal electrode placed in close distance above or next to the cable. In (Theethayi and Thottappillil, 2007), a follow-on bare wire in a horizontal distance of 10 cm was shown to significantly reduce the internal voltages between the core and the cable shield.

In addition to the ideas presented in the preceding paragraph, further investigations are necessary to analyze the impact of the sensor's precise electrical wiring, as this is likely to influence the results in practice. Although not explicitly demonstrated in this study, the simulated shield currents - with the cable shield being disconnected from the ground - yield angle and amplitude site error results that significantly underestimate those occasionally observed in reality when shields are left open-ended. In practice, disconnecting the shield often results in angle site errors reduced to half their original value. This real-world observation could not be fully explained within the scope of the present study. It is hypothesized that, in such cases, additional coupling mechanisms are at play, impacting the site error behavior.

In areas with low local ground conductivities, achieving grounding resistances often recommended by the electrical equipment manufacturers, such as 10  $\Omega$ , is nearly impossible. Instead, grounding resistance tends to increase as the local ground conductivity  $\sigma_{loc}$  decreases. Taking this into account in the site error simulations of the present study yielded results (see Fig. 17 and Fig. 18) that align more closely with the overall behavior of LLS sensor site errors observed by LLS network operators. Interestingly, the most problematic range of local ground conductivities in terms of angle and amplitude site errors lies between  $\sigma_{loc} = 1 \cdot 10^{-3}$  S/m and  $\sigma_{loc} = 10 \cdot 10^{-3}$  S/m, which are commonly found at sensor sites. Thus, a shield connected to ground is typically associated with high site errors, precisely as predicted in Fig. 17 and Fig. 18.

The complex interplay between  $\sigma_p$  (which impacts the rise time of the field),  $\sigma_{loc}$ , peak value ratios  $E_x/E_z$ ,  $E_x(z=-d)/E_x(z=0)$ , the difference between a grounded and a floating cable shield at the sensor end, and their impact on the induced current has been demonstrated theoretically in this study (see Fig. 10a and Fig. 10b). However, it was also emphasized that, even for high local ground conductivity  $\sigma_{loc}$ , burial depth alone does not significantly influence overall site errors. Instead, the increasing vertical distance to the H-field sensor with greater burial depth becomes the dominant factor in reducing the observed site errors. Notably, the exact sensor location plays a crucial role, exhibiting inversely proportional (1/r) site error levels. The higher the sensor is positioned above ground and farther from horizontally buried cable segments, the smaller the sensor site errors. This observation aligns with the experience of LLS network operators.

The seasonal contrast between dry and wet soil due to variations in precipitation and humidity, likely plays a significant role in site errors, as it causes substantial changes in the local ground conductivity. It is well established that the soil conductivity reaches its lowest values during seasons with little rainfall and its highest during periods of frequent rainfall, particularly in the uppermost soil layer (< 1m). This phenomenon is discussed in details in (Coelho et al., 2015).

Moreover, the present study assumes a one-layer ground model. In reality, scenarios are far more complex, often involving stratified ground, inhomogeneous soil (particularly in terms of conductivity, see for instance (Rizki Ramdhani et al., 2020) or (Loke, 2001)), various cables, cable paths, installation circuitry, and diverse grounding methods. Consequently, the theoretical considerations presented in this work, while providing insight into the fundamental principles behind site errors, cannot fully capture the complexity of real-life scenarios. More in-depth investigations, both empirical and theoretical, are left for future research.

#### Conclusion

The presented study constitutes the first attempt to explain the physical mechanisms underlying angle and amplitude site errors when magnetic direction finders (MDF) are employed in lightning location systems (LLSs). From the outset, these errors have been attributed to shield currents in the sensor power supply cable, driven by the horizontal E-field component of the incident lightning EM field, resulting from ground losses. The objective was to present a modeling approach allowing to simulate LLS sensor's angle and amplitude site errors. Specifically, the computational model took into account the whole chain of physical interactions between the lightning EM field and the EM environment during propagation and detection at the sensor site. This process started from the computation of typical lightning EM field incident at the sensor site after propagating over lossy ground. It was followed by determining the horizontal E-fields responsible for driving coupled currents in the sensor power supply cable shield. After theoretically calculating cable shield currents, the resulting scattered magnetic fields, which cause LLS sensor site errors by altering the true incident H-field of interest, were computed using Biot-Savart's law. This involved considering current elements up to 50 meters from the sensor head. By computing the scattered H-fields  $(\vec{H}_{err})$ , it became possible to evaluate the theoretically expected site errors given for a given set of parameters, including the ground conductivity along the propagation path  $\sigma_p$ , the local ground conductivity at the sensor site  $\sigma_{loc}$ , the power supply cable length L, the burial depth d, and the grounding resistance of the shield connected to ground. The applicability and adequacy of each step are supported by a substantial body of literature, cited in this work and providing valuable resources for similar investigations.

The simulations of theoretical scenarios such as insulated and bare single-conductor cables or wires (representing cable shields or grounding electrodes), successfully reproduced angle and amplitude site errors across the entire azimuth range ( $0^{\circ}$ -360°), with satisfactory agreement to real-life observations

| 664        | from operational sensors. The impact of various parameters on the resulting sensor site errors was                                                                                                      |
|------------|---------------------------------------------------------------------------------------------------------------------------------------------------------------------------------------------------------|
| 665        | thoroughly discussed, and key graphs in Fig. 17 and Fig. 18 highlight the influence of the local ground                                                                                                 |
| 666        | conductivity $\sigma_{\text{loc}}$ - and accordingly the grounding resistance - on the maximum expected site errors.                                                                                    |
| 667        | These results provide LLS network operators with a straightforward tool to estimate expected site errors                                                                                                |
| 668        | at provisioned sensor locations, or, retrospectively, to evaluate whether observed site errors align with                                                                                               |
| 669        | expectations.                                                                                                                                                                                           |
| 670        | For optimal behavior, it is recommended that the shield always remain disconnected from ground at the                                                                                                   |
| 671        | sensor-side end, as this minimizes coupled shield currents near the MDF sensor. The observed reduction                                                                                                  |
| 672        | of site errors to approximately half when the shield is disconnected from ground could not be fully                                                                                                     |
| 673        | explained within the scope of this study, requiring further in-depth investigations. Furthermore, bare                                                                                                  |
| 674        | wires (e.g., horizontal ground electrodes) exhibit smaller site errors and show a significantly reduced                                                                                                 |
| 675        | dependency on wire length. Thus, they can be beneficial as follow-on electrodes parallel to the supply                                                                                                  |
| 676        | cables to reduce site errors.                                                                                                                                                                           |
| 677        | The simulations also replicated subtle deviations from a perfectly symmetric double-cycle sinusoidal                                                                                                    |
| 678        | site error behavior. These nuances, apparent when comparing Fig. 2 and Fig. 15, and further confirmed                                                                                                   |
| 679        | by Fig. 19, corroborate the reliability of the study's results.                                                                                                                                         |
| 680        | The presented methodology provides a solid foundation for further studies related to supply cable-                                                                                                      |
| 681        | induced LLS sensor site errors. Subsequent investigations should aim to identify optimal configuration                                                                                                  |
| 682        | for LLS sensors at specific sensor site locations.                                                                                                                                                      |
| 683        | Code/Data availability:                                                                                                                                                                                 |
| 684        | The code and data can be provided upon request at OVE Service GmbH (corresponding author)                                                                                                               |
| 685        | Competing interests:                                                                                                                                                                                    |
| 686        | The authors declare that they have no conflict of interest                                                                                                                                              |
| 687        | Author contribution:                                                                                                                                                                                    |
| 688        | HK: Methodology, Simulation, Results, manuscript writing                                                                                                                                                |
| 689        | WS: LLS sensor data extraction and preparation, manuscript writing, proof reading                                                                                                                       |
| 690        | FR: Proof reading, contextual and editorial corrections and suggestions                                                                                                                                 |
| 691        | ND: Assistance and validation in ground modeling and field-to-cable coupling                                                                                                                            |
| 692        | DK: Result validation of the simulations, assistance w.r.t. the FDTD-method                                                                                                                             |
| 693        | Acknowledgements:                                                                                                                                                                                       |
| 694<br>695 | The authors wish to thank Prof. Rafael Alipio from the Federal Center for Technological Education of Minas Gerais (CEFET-MG) for sharing his valuable expertise during insightful discussions regarding |

field-to-cable coupling and the frequency-dependent wave propagation phenomena in soil. Further the

696 697

- (AFSCET) for sharing his thoughts on potential interactions in terms of EMC in the domain of LLS
- sensors, therefore adding interesting material for further in-depth studies.

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
