# Peer review of "Analysis of Lightning-Induced Currents in Supply Cable Shields and Their Impact on LLS Sensor Site Errors"

_EGUsphere, 2025_

## Referee Comment (RC1)

Review of "Analysis of Lightning-Induced Currents in Supply Cable Shields and Their Impact on LLS Sensor Site Errors " by Kohlmann et al., initial draft

Reviewer: Martin Murphy

The paper presents a very interesting "deep dive" into modeling the sources of angle and amplitude site errors at sensor sites in lightning locating systems. The modeling work here provides useful guidance in planning the layout of sensor sites with the goal of minimizing the errors in the first place. The paper is in good shape overall, although I do have some suggestions about clarifications, as follows:

equation 3 looks like it may have an error – it should have a "d" somewhere in it; otherwise, the exponential term is just a constant

line 214 vs lines 226-227: $Y_g$ is referred to as "impedance" in the first place, and "admittance" farther below.

lines 267-268: "considering a channel-base current typical of subsequent return strokes, as depicted in Fig. 4" - should we assume that all calculations, even as far down as figures 11, 12, etc, are all done using a stroke peak current of 12 kA, given the words here about "as depicted in fig. 4"? Or does the stroke peak current ever vary in the calculations presented farther down?

line 298: "ground conductivity values ranging from $\sigma_p = 10^{-1}$ S/m to $\sigma_p = 10^{-4}$ mS/m" That second unit of measurement should be S/m, I think, rather than mS/m. More generally, it might just make more sense to stick with one unit of measurement, whether mS/m or S/m, throughout the paper.

lines 299-300: "(1) fields with shorter rise times (fast transients) tend to create larger $E_x$-field peaks, and (2) fast transients are better preserved over propagation path with high conductivity σp" – I think that I may have lost touch a bit: in figure 10a, I assume that the frequency content of the lightning signal is an issue _indirectly_, via the fact that higher values of $\sigma_p$ attenuate the high-frequency content of the original signal less, as shown in figure 7 rather than figure 10a. Is my understanding correct, or have I missed something?

lines 365-368: it is worth pointing out that the sampling instant in question here applies only to the measurement of angle of arrival and peak amplitude, but the sampling of the arrival time is hardly affected because times of arrival are measured as close as possible to the start of the rising edge of the waveform, precisely to avoid the significant delay of the peak due to propagation effects.

lines 404-414: discussion surrounding figure 15 appears to be on solid ground, but slightly confusing. In lines 407-408, "For a burial depth of 1.5 m, the angle site errors $\alpha_{err}$ decrease by only -8.5%, while the total reduction reaches -46%" you may want to clarify that the decrease of 8.5% addresses only the cable depth component, whereas the term "total reduction" is the combination of cable depth plus increased distance to the sensing antenna when the antenna is kept at 2 m above ground. It is also not exactly clear what is meant by "Thus, the contribution of the cable distance to the sensor remained practically the same, as expected" at the end of that section: In figure 15a, the combined total reduction actually appears to be about 3 degrees zero to peak, as opposed to the 1.3-degree reduction (3.07 vs 1.78) stated in the high-conductivity case.

lines 470-489 make reference to "wave propagation effects" several times. This may be another place where I've lost touch with earlier sections of the paper. I see "wave propagation" in line 163,

where it clearly appears to refer to the effects on the overall signal as it propagates long distances over lossy ground. Then again "wave propagation" appears in line 373, which is a reference to the vertical penetration of the $E_x$ component and thus the induction of current on the cable. In lines 470-483, I think that the "wave propagation effects" refer to the vertical penetration part, but it's not entirely clear, at least not to me.

---

## Referee Comment (RC2)

Comments on egusphere 2025-1015

Title: Analysis of Lightning-Induced Currents in Supply Cable Shields and Their Impact on LLS Sensor Site Errors

Author(s): Hannes Kohlmann et al.

Reviewer: Kenneth Cummins

**General Comments**

This work is a quantitative, detailed effort to explain the physical mechanisms contributing to site errors in magnetic direction finders used to locate lightning. The authors provide a clear, logical, and well-organized presentation of the physical "steps" leading to the unwanted magnetic field seen by a near-ground sensor that is produced by buried cables associated with the sensor. Briefly, these steps include propagation of the lightning-produced EM signal over lossy ground, coupling of the resulting field onto the buried cables producing current in the cables, and the resulting magnetic field produced by this cable current and sensed by the instrument. This all leads to angle-dependent errors in both the magnitude and inferred direction of the observed lightning magnetic field wavefront.

The authors model all the steps in this process and evaluate their behavior in terms of the physical properties that influence site errors. The cable properties include cable length, depth, diameter, electrical properties, and grounding methods. Ground electrical conductivity is separately parameterized for the propagation path from the lightning discharge to the sensor, and for the local conditions at the site. The geometrical relationship between the conductors (assumed to terminate at the sensor) and the sensing loops of the sensor is parameterized in terms of separation distance and conductor depth in the ground. All-in-all, this is the first in-depth exploration of cable-caused site errors that this reviewer has seen, and it is a clear and poignant scientific and technical contribution for those who strive to understand or improve the performance of lightning locating systems employing magnetic direction finding.

I have no major concerns/issues with the approach taken in this work or with the findings, but I do have some questions, comments and "issues" that I would like to convey to the authors. I also have a number of minor and editorial comments and suggestions. These are enumerated in separate sections below. I have seen the comments by the other reviewer (Martin Murphy), so I do not include issues that he identified.

**Scientific/Technical Questions/Comments/Issues**

1. One over-arching issue for this reviewer is the number of analyses that were carried-out before providing results for a simple "bare wire" case in Fig. 14. It is my understanding that that it is common practice to place a bare wire 20-30 cm directly above the insulated power and communications cables, to serve as lightning protection for these cables. Given the clear benefit of this for also reducing site errors, it was surprising that the analyses for field-to-cable coupling and scattered field values did not include this condition. The authors may have reasons for this.

2. A question for the authors: do you know if the case of a bare wire above the insulated cables will produce the same coupling behaviors as a bare wire by itself? I am not sure about this.

3. The stimulating content in this study has caused me to think about unusual geometries for the underground cables to the sensors. What about a long cable coming towards the sensor, but offset laterally by about 6-8m. Then, at the appropriate distance, the cables could make a ~90-degree turn to go to the sensor? The long cable would be far enough away to produce an minimal site error (see your Fig. 16), and then the short cable near the sensor would have much less coupled current and its site-error would be out-of-phase with the long cable's contribution. Comments?

4. Frequent use of 0.001 and 0.0001 S/m conductivity may not align well with typical LLS locations. Starting with Fig. 7, many of the simulations employ this very-low local near-surface electrical conductivity. It would be appropriate to justify this early in the manuscript. In my experience, this value only exists for dry or non-porous rock.

5. The simulations typically used a fixed propagation distance and then varied the path electrical conductivity. It might be helpful to state that for a path with fairly uniform electrical conductivity, shortening the path length by some percentage is approximately the same as decreasing the electrical conductivity in the same proportion. SO – nearby lightning will have more high-frequency content than distant lightning. This should therefore change the site error magnitude as a function of distance, given the right mix of conditions. Might this be wort stating?

6. I am confused about Fig. 11, although I may have figured out some of the issues. In both panels in Fig. 11, the legends say "L = nnn", but I think that it should be "x = nnn". If these are really positions along the line, then it should be stated in the caption or the body of the manuscript. Also, the text states that the peak current at the line end in Fig. 11(a) is 82 mA, but it looks like 100 mA. For both Fig. 11 a &b, I am unable to reconcile the 15-20 microsecond periodicity in the current waveforms. Why does this periodicity exist for the cable grounded at both ends (11a)? This period also seems quite long, given that the round-trip time to one end and back, at the speed of light, is about 3 microseconds. A discussion of this would be helpful.

7. It seems that there is a difference between the insulted wire current waveforms for L=100 in Fig. 12 (red waveform) and Fig. 14a (blue waveform). The amplitudes, fall time, and subtitles in the shape all differ.

8. Figures 17 &18 are very nice illustrations, but the parameter domains may not be ideal. The "short" risetime associated with 0.001 S/m does not provide the likely subsequent stroke risetimes seen at 100-200 km, and the "long" risetime case seems unrealistic. Also, inclusion of 0.00001 S/m in the domain for the local conductivity is probably unnecessary, and produces unstable behaviors.

**Minor/Editorial Comments/Suggestions**

9. Line 29: suggest changing "are network of sensors" to "include a network of sensors", since an LLS is more than just sensors.
10. Line 33: misspelling of "Cooray"
11. Line 35: (picky point) - I note that Vaisala's IMPACT LLS's use the absolute time of arrival, relative to the estimated discharge time, rather than time-differences between sensors.
12. Line 48: suggest adding "additive" after "spurious"
13. Lines 49-50: suggest eliminating "at the sampling instants", since the spurious fields superimpose on the whole waveform.
14. Line 54: suggest adding "nearby" before "ground"
15. Line 69: suggest changing "serves as an estimator for" to "is used to produce the estimated"
16. Line 72: The second half of this line, starting with "respectively", does not seem to fit here.

17. Line 105: suggest changing "shield currents" to "cable currents" or something like this, since a bare wire is one of the conditions.
18. Line 108: suggest changing "MDF techniques to locate lightning." to "magnetic fields to locate lightning and/or estimate peak current."
19. Line 112: suggest adding "cable grounding method" to the list
20. Line 249: suggest changing "introduces an error to" to "adds a spurious term beyond", or something like this, since it does not change the true incident field.
21. Line 326: unnecessary line break after "shows"
22. Line 329: should this be "line end", and not "cable end"?
23. Line 330: Should the percentage be 60%, rather than 75%?
24. Lines 336-7: suggest changing "local conductivity on the site reduces" to "local conductivity is lower"
25. Line 351: suggest changing "distance of the MDF to" to "vertical separation between the MDF antennae and"
26. Fig. 15: I cannot find the cable length that was used for this study
27. (suggestions beyond Line 351 have not been transcribed /refined yet)
28.

---

## Referee Comment (RC3)

Comments on egusphere-2025-1015

Title: Analysis of Lightning-Induced Currents in Supply Cable Shields and Their Impact on LLS Sensor Site Errors

Author(s): Kohlmann et al.

Reviewer: Dylan Goldberg

**General Comments**

This study provides an interesting and thorough analysis on the physical mechanisms behind angle and amplitude site errors for magnetic direction finders in lightning location systems. This work is important for understanding the potential errors due to sensor and cable placement associated with sensor site locations. The authors provide a clear and concise description of their methodology towards evaluating sensor site errors and determining which variables are most important to their analysis. I don't have any major issues with this work but I have a few minor comments to add without repeating what the other reviewers have suggested.

**Specific/Technical Comments**

- Line 114: Is the theoretical shielded conductor single or double shielded? Do the results of this study apply to both? It would also be interesting for a future study to see if the results vary between types of shielding used (i.e. braid vs foil).
- Lines 204 - 205: Should "finite-length" be removed since that (at least to me) is implied by a cable of length L?
- Lines 244 - 245: It's stated that contributions beyond 50 m are assumed to be negligible. Was this number decided based on a percentage from the $1/r^2$ dependency or were contributions beyond this number tested initially?
- Figure 7: The ground electric conductivity for the red line should also have units in the legend to be consistent with the other lines.
- Figure 8: The vertical axis labels are missing the closing parenthesis around the units. Also, should the d = -0.0 m burial depth be negative?
- Figure 9: It may be beneficial to use the same notation for the conductivities across the different figures and text; maybe switch the legend conductivities from scientific notation to magnitudes as shown in the figure caption? Also, I'm not sure I understand the change in the time scale of the horizontal axis for this figure compared to figures 7 and 8.
- Line 308: The equation "d=1m" should probably be "d = 1 m" (spaces added) to be consistent with the other inline equations. This also occurs in the legends of some figures.

- Line 326 - 327: There seems to be an extra line/paragraph starting here.
- Figure 19: Fig (a) appears to be lower in resolution (dpi) quality than the other figures in this paper.

---

## Referee Comment (RC4)

Comments on egusphere 2025-1015

Title: Analysis of Lightning-Induced Currents in Supply Cable Shields and Their Impact on LLS Sensor Site Errors

Author(s): Hannes Kohlmann et al.

Reviewer: Kenneth Cummins

**General Comments**

This work is a quantitative, detailed effort to explain the physical mechanisms contributing to site errors in magnetic direction finders used to locate lightning. The authors provide a clear, logical, and well-organized presentation of the physical "steps" leading to the unwanted magnetic field seen by a near-ground sensor that is produced by buried cables associated with the sensor. Briefly, these steps include propagation of the lightning-produced EM signal over lossy ground, coupling of the resulting field onto the buried cables producing current in the cables, and the resulting magnetic field produced by this cable current and sensed by the instrument. This all leads to angle-dependent errors in both the magnitude and inferred direction of the observed lightning magnetic field wavefront.

The authors model all the steps in this process and evaluate their behavior in terms of the physical properties that influence site errors. The cable properties include cable length, depth, diameter, electrical properties, and grounding methods. Ground electrical conductivity is separately parameterized for the propagation path from the lightning discharge to the sensor, and for the local conditions at the site. The geometrical relationship between the conductors (assumed to terminate at the sensor) and the sensing loops of the sensor is parameterized in terms of separation distance and conductor depth in the ground. All-in-all, this is the first in-depth exploration of cable-caused site errors that this reviewer has seen, and it is a clear and poignant scientific and technical contribution for those who strive to understand or improve the performance of lightning locating systems employing magnetic direction finding.

I have no major concerns/issues with the approach taken in this work or with the findings, but I do have some questions, comments and "issues" that I would like to convey to the authors. I also have a number of minor and editorial comments and suggestions. These are enumerated in separate sections below. I have seen the comments by the other reviewer (Martin Murphy), so I do not include issues that he identified.

**Scientific/Technical Questions/Comments/Issues**

1. One over-arching issue for this reviewer is the number of analyses that were carried-out before providing results for a simple "bare wire" case in Fig. 14. It is my understanding that that it is common practice to place a bare wire 20-30 cm directly above the insulated power and communications cables, to serve as lightning protection for these cables. Given the clear benefit of this for also reducing site errors, it was surprising that the analyses for field-to-cable coupling and scattered field values did not include this condition. The authors may have reasons for this.

2. A question for the authors: do you know if the case of a bare wire above the insulated cables will produce the same coupling behaviors as a bare wire by itself? I am not sure about this.

3. The stimulating content in this study has caused me to think about unusual geometries for the underground cables to the sensors. What about a long cable coming towards the sensor, but offset laterally by about 6-8m. Then, at the appropriate distance, the cables could make a ~90-degree turn to go to the sensor? The long cable would be far enough away to produce an minimal site error (see your Fig. 16), and then the short cable near the sensor would have much less coupled current and its site-error would be out-of-phase with the long cable's contribution. Comments?

4. Frequent use of 0.001 and 0.0001 S/m conductivity may not align well with typical LLS locations. Starting with Fig. 7, many of the simulations employ this very-low local near-surface electrical conductivity. It would be appropriate to justify this early in the manuscript. In my experience, this value only exists for dry or non-porous rock.

5. The simulations typically used a fixed propagation distance and then varied the path electrical conductivity. It might be helpful to state that for a path with fairly uniform electrical conductivity, shortening the path length by some percentage is approximately the same as decreasing the electrical conductivity in the same proportion. SO – nearby lightning will have more high-frequency content than distant lightning. This should therefore change the site error magnitude as a function of distance, given the right mix of conditions. Might this be wort stating?

6. I am confused about Fig. 11, although I may have figured out some of the issues. In both panels in Fig. 11, the legends say "L = nnn", but I think that it should be "x = nnn". If these are really positions along the line, then it should be stated in the caption or the body of the manuscript. Also, the text states that the peak current at the line end in Fig. 11(a) is 82 mA, but it looks like 100 mA. For both Fig. 11 a &b, I am unable to reconcile the 15-20 microsecond periodicity in the current waveforms. Why does this periodicity exist for the cable grounded at both ends (11a)? This period also seems quite long, given that the round-trip time to one end and back, at the speed of light, is about 3 microseconds. A discussion of this would be helpful.

7. It seems that there is a difference between the insulted wire current waveforms for L=100 in Fig. 12 (red waveform) and Fig. 14a (blue waveform). The amplitudes, fall time, and subtitles in the shape all differ.

8. Figures 17 &18 are very nice illustrations, but the parameter domains may not be ideal. The "short" risetime associated with 0.001 S/m does not provide the likely subsequent stroke risetimes seen at 100-200 km, and the "long" risetime case seems unrealistic. Also, inclusion of 0.00001 S/m in the domain for the local conductivity is probably unnecessary, and produces unstable behaviors.

**Minor/Editorial Comments/Suggestions**

9. Line 29: suggest changing "are network of sensors" to "include a network of sensors", since an LLS is more than just sensors.
10. Line 33: misspelling of "Cooray"
11. Line 35: (picky point) - I note that Vaisala's IMPACT LLS's use the absolute time of arrival, relative to the estimated discharge time, rather than time-differences between sensors.
12. Line 48: suggest adding "additive" after "spurious"
13. Lines 49-50: suggest eliminating "at the sampling instants", since the spurious fields superimpose on the whole waveform.
14. Line 54: suggest adding "nearby" before "ground"
15. Line 69: suggest changing "serves as an estimator for" to "is used to produce the estimated"
16. Line 72: The second half of this line, starting with "respectively", does not seem to fit here.

17. Line 105: suggest changing "shield currents" to "cable currents" or something like this, since a bare wire is one of the conditions.

18. Line 108: suggest changing "MDF techniques to locate lightning." to "magnetic fields to locate lightning and/or estimate peak current."

19. Line 112: suggest adding "cable grounding method" to the list

20. Line 249: suggest changing "introduces an error to" to "adds a spurious term beyond", or something like this, since it does not change the true incident field.

21. Line 326: unnecessary line break after "shows"

22. Line 329: should this be "line end", and not "cable end" ?

23. Line 330: Should the percentage be 60%, rather than 75%?

24. Lines 336-7: suggest changing "local conductivity on the site reduces" to "local conductivity is lower"

25. Line 351: suggest changing "distance of the MDF to" to "vertical separation between the MDF antennae and"

26. Fig. 15: I cannot find the cable length that was used for this study

27. Caption for Fig. 13: The first line of the caption says "site errors", but it is actually the magnetic fields

28. Line 364: might want to say "contributes to H_sampled" rather than H_y, since the scattered field does not change H_y

29. Lines 377 to 381: This might be a good place to note that there will also be high frequency content in the incident field when the propagation distance is short, as long as the sensor bandwidth accommodates the higher frequencies.

30. Line 390: suggest changing "such as" to "including", since this is what you have done.

31. Line 416: The text states "distance to the cable", but it would be more precise to say "vertical distance to the cable." This wording occurs in other places in the manuscript

32. Line 434: It would be clearer to say "propagation path ground conductivity" instead of "ground parameters"

33. Line 443: does this simply repeat what is stated at the end of the previous paragraph?

34. 467-469: Does this statement also appyl to the bare conductor case?

35. Line 486: suggest adding "local" before "ground" (also in other places in the manuscript)

36. Line 487: suggest changing "left to the peak" to "left of the peak" or "before the peak"

37. Line 545: should probably say "above the cable", rather than "from the cable". This addresses my Comment 1 above! Might not be needed if you know the answer to my Comment 2.
38. Line 556: please add "local" before "conductivities"
39. Lines 562-3: should this be clear about the difference between grounding at both ends, vs. just grounding at the end that is far from the sensor? You discuss this at Line 613-615, but it is not made clear as part of the study.

End

---

## Author Comment (AC1)

Review of "Analysis of Lightning-Induced Currents in Supply Cable Shields and Their Impact on LLS Sensor Site Errors " by Kohlmann et al., initial draft Reviewer: Martin Murphy

**The authors would like to thank Martin Murphy for his valuable review, positive remarks and constructive feedback. His comments and questions allowed to improve the paper.**

The paper presents a very interesting "deep dive" into modeling the sources of angle and amplitude site errors at sensor sites in lightning locating systems. The modeling work here provides useful guidance in planning the layout of sensor sites with the goal of minimizing the errors in the first place. The paper is in good shape overall, although I do have some suggestions about clarifications, as follows:

equation 3 looks like it may have an error – it should have a "d" somewhere in it; otherwise, the exponential term is just a constant.

Indeed, thank you for the hint. The equation was corrected and was formulated more generally in terms of a depth "z".

line 214 vs lines 226-227: Yg is referred to as "impedance" in the first place, and "admittance" farther below.

Yes, it should have read 'admittance'. Corrected!

lines 267-268: "considering a channel-base current typical of subsequent return strokes, as depicted in Fig. 4" - should we assume that all calculations, even as far down as figures 11, 12, etc, are all done using a stroke peak current of 12 kA, given the words here about "as depicted in fig. 4"? Or does the stroke peak current ever vary in the calculations presented farther down?

Due to the linearity of the utilized equations throughout the whole paper, the peak current can be chosen arbitrarily. It could have been normalized to 1 (k)A. We have chosen this waveform because subsequent return stroke currents exhibit higher frequency content compared to first return strokes. The waveform exhibits a short rise time which can be affected by the ground parameters along the propagation path to obtain the fields at the sensor site that exhibit the desired characteristics. Scaling the channel-base current amplitude by a constant factor (for example in order to obtain different current- or field peaks) would not have affected the presented results based on ratios (i.e., angle and amplitude site errors). Figures showing absolute values for results of field and current (i.e., non-ratio values) are associated with the respective fields, as indicated in the figure captions.

   The segment around lines 267 and 268 was adapted to point out more clearly that the amplitude was not varied throughout the paper:

*This section presents the simulation results of lightning incident electric fields following the procedure described in Section 2.1, considering a channel-base current waveform that exhibits characteristics that are typical of subsequent return strokes (in particular, characterized by a short risetime), as depicted in Fig. 4. All results are obtained for a distance to the lightning discharge of 100 km. Due to the linearity and time-invariance of the equations utilized in this paper, the amplitude of the channel-base current was kept constant throughout all computations. Variations of the E-fields used as input for the coupling analyses were solely the result of the assumed ground parameters along the propagation path (see Fig. 7). The main results of this paper, namely the angle and amplitude site errors, are independent of the selected channel-base current amplitude; that is they are unaffected by any scaling of the waveform.*

line 298: "ground conductivity values ranging from $\sigma_p = 10^{-1}$ S/m to $\sigma_p = 10^{-4}$ mS/m" That second unit of measurement should be S/m, I think, rather than mS/m. More generally, it might just make more sense to stick with one unit of measurement, whether mS/m or S/m, throughout the paper.
Thank you for the correction and suggestion. The unit of measurement has been standardized to S/m throughout the paper. To better highlight differences among several orders of magnitude, the values are now expressed using a common factor of $10^{-3}$, replacing the previous use of "m". Ground

conductivities are therefore presented as coefficient multiplied by $10^{-3}$ S/m, for example: $50 \cdot 10^{-3}$ S/m, or $1 \cdot 10^{-3}$ S/m, or $0.01 \cdot 10^{-3}$ S/m.

lines 299-300: "(1) fields with shorter rise times (fast transients) tend to create larger Ex-field peaks, and (2) fast transients are better preserved over propagation path with high conductivity σp" – I think that I may have lost touch a bit: in figure 10a, I assume that the frequency content of the lightning signal is an issue indirectly, via the fact that higher values of σp attenuate the high-frequency content of the original signal less, as shown in figure 7 rather than figure 10a. Is my understanding correct, or have I missed something?

You understood it perfectly right! Thank you for pointing out the ambiguity of the sentence, it was indeed presenting the key aspect of Fig. 10 in a confusing way. Some information was added and some other info removed to that paragraph in order to point out the results of Fig. 10(a) more clearly:

*As previously shown in Fig. 7, propagation over a highly conducting ground (ideally PEC) preserves the high-frequency content of the propagating fields. This results in incident fields exhibiting fast transients and corresponding short risetimes. In contrast, propagation over less conductive ground attenuates the high-frequency content and causes dispersion, leading to incident fields with slower transients and longer risetimes. Examination of Fig. 10a now reveals two key aspects. (1) fields with shorter rise times (fast transients) produce larger $E_x$-field peaks (as evidenced by the bold blue curve with the thin red curve at a given local ground conductivity $\sigma_{loc}$) and (2) low local ground conductivity produces large $E_x$-field peaks, whereas highly conductive local ground reduces the $E_x$-field peak significantly that eventually reaches zero for infinite ground conductivity $\sigma_{loc}$ (PEC ground). A realistic scenario for a lightning EM field involves propagation over lossy ground with conductivity values $\sigma_p$ between $0.1 \cdot 10^{-3}$ S/m and $10 \cdot 10^{-3}$ S/m over 100 km, resulting in incident fields similar to those shown in Fig. 7.*

lines 365-368: it is worth pointing out that the sampling instant in question here applies only to the measurement of angle of arrival and peak amplitude, but the sampling of the arrival time is hardly affected because times of arrival are measured as close as possible to the start of the rising edge of the waveform, precisely to avoid the significant delay of the peak due to propagation effects.

Thank you for that comment. The following note was inserted between lines 366 and 367:

*Note that the estimated time of arrival is not significantly affected by the addition of the $\vec{H}_{err}$ field, as it is determined as close as possible to the onset of the waveform's rising edge. Thus, the LLS location results obtained using the ToA technique remain unaffected by the phenomena illustrated in Fig. 13.*

lines 404-414: discussion surrounding figure 15 appears to be on solid ground, but slightly confusing. In lines 407-408, "For a burial depth of 1.5 m, the angle site errors αerr decrease by only -8.5%, while the total reduction reaches -46%" you may want to clarify that the decrease of 8.5% addresses only the cable depth component, whereas the term "total reduction" is the combination of cable depth plus increased distance to the sensing antenna when the antenna is kept at 2 m above ground. It is also not exactly clear what is meant by "Thus, the contribution of the cable distance to the sensor remained practically the same, as expected" at the end of that section: In figure 15a, the combined total reduction actually appears to be about 3 degrees zero to peak, as opposed to the 1.3-degree reduction (3.07 vs 1.78) stated in the high-conductivity case.

Thank you for the comment. The paragraph was now split up into two parts, giving more explanation and reasoning behind the idea of recalculating the impact of the burial depth for a higher ground conductivity. Further, Scenario 1 (combined effect of ground attenuation + distance between the cable and the sensor) and Scenario 2 (accounting merely for the ground attenuation) are more

explicitly described further above. The whole segment, including the bullet points of scenarios 1 and 2, now reads as follows, should now allow for a better reading flow and be much better interpretable:

*We begin by examining the impact of the burial depth of the power supply cable on the site errors. The simulation results are presented in Fig. 15 and cover two distinct scenarios:*

- *Scenario 1: As the burial depth increases, the distance between the cable to the sensor head also increases, reflecting the most realistic scenario. In this case, the site error reduction is influenced by a combined effect of increasing distance between the cable to the H-field sensor and the field attenuation by the ground (solid lines in Fig. 15).*
- *Scenario 2: The cable is buried at different depths, but the relative distance between the cable and the H-field sensor is kept constant at 2 meters. This scenario isolates the effect of ground attenuation from the distance effect, highlighting their distinct contribution. The impact of ground attenuation alone is shown in dashed lines in Fig. 15.*

*The results presented in Fig. 15 were obtained for a local ground conductivity $\sigma_{loc} = 10 \cdot 10^{-3}$ S/m. They reveal a significant finding: The site errors are very strongly impacted by the (vertical) distance of the cable to the H-field sensor, as indicated by the solid-line curves. In contrast, the dashed-line curves, representing the scenario with a fixed 2-m distance, exhibit only a minor reduction in site errors with increasing burial depth. Specifically, at a burial depth of 1.5 m in Scenario 2, the angle site error $\alpha_{err}$ is reduced by only 8.5%. However, in Scenario 1, where the cable-to-sensor distance increases with burial depth, the reduction reaches 46%. This finding is consistent with results presented in Fig. 10b which suggests the same effect based on the attenuation caused by the ground penetration of the $E_x$-field for the assumed parameters. The amplitude site errors $s_{err}$ exhibit a similar trend, decreasing by comparable amounts.*

*Next, the impact of a significantly higher local ground conductivity $\sigma_{loc}$ is investigated. As shown previously in Fig. 10b, higher conductivity increases the attenuation of the illuminating Ex-field as it penetrates to ground. Additionally, Fig. 10a demonstrated that higher $\sigma_{loc}$ leads to smaller site errors due to the reduced horizontal Ex-field illuminating the cable shield. To account for this effect, a new baseline angle site error was calculated for a cable placed at ground level (d = 0 m) and a sensor located 2 m above, assuming a value for the local ground conductivity of $\sigma_{loc} = 50 \cdot 10^{-3}$ S/m. The angle site error in this case drops to 3.86°, compared to 7.5° for $\sigma_{loc} = 10 \cdot 10^{-3}$ S/m at an azimuth of 130°, for example. Using this new baseline angle site error, the impact of ground attenuation for a buried cable is re-evaluated. For Scenario 2 (only the effect of ground attenuation), the angle site error is reduced by 20% at a burial depth of d = 1.5 m, compared to just 8.5% for the lower conductivity case $\sigma_{loc} = 10 \cdot 10^{-3}$ S/m. In Scenario 1 (which includes both ground attenuation and increased distance to the sensor), the reduction reaches 54%, compared to 46% for $\sigma_{loc} = 10 \cdot 10^{-3}$ S/m.*

*Thus, while the attenuation-caused reduction is greater for higher $\sigma_{loc}$ (20% vs. 8.5%), the dominant factor contributing to the total site error reduction in Scenario 1 remains the increased vertical distance between the sensor and the cable. It is important to note that these findings are independent of the significant overall decrease in site error of almost 50% (for $\sigma_{loc} = 50 \cdot 10^{-3}$ S/m in contrast to $\sigma_{loc} = 10 \cdot 10^{-3}$ S/m) that results directly from the reduced Ex-field strength at high local ground conductivity.*

lines 470-489 make reference to "wave propagation effects" several times. This may be another place where I've lost touch with earlier sections of the paper. I see "wave propagation" in line 163, where it clearly appears to refer to the effects on the overall signal as it propagates long distances over lossy ground. Then again "wave propagation" appears in line 373, which is a reference to the vertical penetration of the Ex component and thus the induction of current on the cable. In lines 470-483, I think that the "wave propagation effects" refer to the vertical penetration part, but it's not entirely clear, at least not to me.

Thank you for the input. Indeed, the wave propagation effects mentioned in 163 with regard to Wait's work is related to the propagation of electromagnetic fields (of arbitrary kind & polarization). The "wave propagation effects" appearing in line 373, in turn, relate to the *''propagation effects of the*

*induced cable shield current wave" (*which result to pronounced reflections and resonances along long lines). *The whole sentence was now extended to highlight that fact:*

*However, for very low ground conductivities (0.1·10$^{-3}$ S/m and below, see Fig. 13c Fig. 13d), the induced current wave on the cable shield experiences minimal attenuation as it propagates along the shield. This leads to  pronounced reflections and resonances along long lines.*

The "wave propagation" occurring between lines 470 and 483 towards the end of the paper again relate to the coupled shield currents that propagate along the line as a travelling wave. This lack of information was addressed by adding the information to "wave propagation": *" propagation of the induced current wave on the cable shield".*

---

## Author Comment (AC2)

**General Comments**

This work is a quantitative, detailed effort to explain the physical mechanisms contributing to site errors in magnetic direction finders used to locate lightning. The authors provide a clear, logical, and well-organized presentation of the physical "steps" leading to the unwanted magnetic field seen by a near-ground sensor that is produced by buried cables associated with the sensor. Briefly, these steps include propagation of the lightning-produced EM signal over lossy ground, coupling of the resulting field onto the buried cables producing current in the cables, and the resulting magnetic field produced by this cable current and sensed by the instrument. This all leads to angle-dependent errors in both the magnitude and inferred direction of the observed lightning magnetic field wavefront.

The authors model all the steps in this process and evaluate their behavior in terms of the physical properties that influence site errors. The cable properties include cable length, depth, diameter, electrical properties, and grounding methods. Ground electrical conductivity is separately parameterized for the propagation path from the lightning discharge to the sensor, and for the local conditions at the site. The geometrical relationship between the conductors (assumed to terminate at the sensor) and the sensing loops of the sensor is parameterized in terms of separation distance and conductor depth in the ground. All-in-all, this is the first in-depth exploration of cable-caused site errors that this reviewer has seen, and it is a clear and poignant scientific and technical contribution for those who strive to understand or improve the performance of lightning locating systems employing magnetic direction finding.

I have no major concerns/issues with the approach taken in this work or with the findings, but I do have some questions, comments and "issues" that I would like to convey to the authors. I also have a number of minor and editorial comments and suggestions. These are enumerated in separate sections below. I have seen the comments by the other reviewer (Martin Murphy), so I do not include issues that he identified.

The authors are thankful for the very positive first assessment. In particular, they wish to thank Kenneth Cummins for the meticulous inspection and the careful listing of the issues and comments. Based on this valuable input, the manuscript has been improved and re-examined once more with respect to the plausibility of individual steps, the selection of parameters, and the overall rigor of the methodology.

**Scientific/Technical Questions/Comments/Issues**

1. One over-arching issue for this reviewer is the number of analyses that were carried-out before providing results for a simple "bare wire" case in Fig. 14. It is my understanding that that it is common practice to place a bare wire 20-30 cm directly above the insulated power and communications cables, to serve as lightning protection for these cables. Given the clear benefit of this for also reducing site errors, it was surprising that the analyses for field-to-cable coupling and scattered field values did not include this condition. The authors may have reasons for this.

Indeed, this practice was not always followed. Initially, it was considered for many of the sensor supplies, but after re-locating sensors, it may have been omitted due to time constraints, cost reduction, or simply for the sake of simplicity. Further, the most interesting sensor investigated for its site-error behavior (see Fig. 19) also does not have a bare wire above it. The authors were aware from the outset, that even the very simple scenario of induced currents and site-errors for a straight sensor supply cable, whose diameter is assumed according to the shield diameter including the jacket (insulation), respectively a simple bare-wire in comparison, would already constitute a very extensive study. More complicated scenarios, such as the one with a protective conductor on top of an insulated

supply cable, would have made the paper significantly longer. This is because additional factors would have to be considered, such as the coupling between the bare conductor and the power supply cable (shield), the terminating conditions, and the resulting reflections due to additional current paths. However, as the impact on the results is of considerable interest, this scenario is planned to be treated in a future study.

2. A question for the authors: do you know if the case of a bare wire above the insulated cables will produce the same coupling behaviors as a bare wire by itself? I am not sure about this.

This is a very interesting question, which will definitely be addressed in a future study due to its pertinence in the application. However, as stated in our answer to your Question 1, this was not considered in the simulations yet. Thus, we are not able to give a definitive answer to this question at this time. Since the present study demonstrated that the overall simulation methodology is capable of reproducing site errors in very good agreement with reality for a sensor that is very specific in its installation (sensor supplied by a 600-m long supply cable in open terrain with a grounded cable shield at the sensor-side), a future study can focus on the details of specific installation schemes. These may include configurations with a protective bare wire of various lengths, nearby metallic structures, or additional supply cables of various lengths, as well as more complex real-life scenarios such as supply cable paths deviating from a purely straight geometry (see Question 3).

3. The stimulating content in this study has caused me to think about unusual geometries for the underground cables to the sensors. What about a long cable coming towards the sensor, but offset laterally by about 6-8m. Then, at the appropriate distance, the cables could make a ~90-degree turn to go to the sensor? The long cable would be far enough away to produce a minimal site error (see your Fig. 16), and then the short cable near the sensor would have much less coupled current and its site-error would be out-of-phase with the long cable's contribution. Comments?

Excellent question. This is actually a very common sensor-supply scenario. In one case in Austria, a sensor was re-located in a way that the supply cable made a 90° turn and had an even larger offset from the longer part of the cable than the 6-8 m discussed here. This led to significantly higher site errors (according to Wolfgang Schulz). The simulation of such a scenario will need to take into consideration: the additional coupling of the short cable segment (in addition to the induction along the long part of the cable), linear superposition of all individual induced current components, the mutual inductance near the 90° turn, etc., and the contribution of all current elements to the magnetic field at the sensor head according to Biot-Savart's law. Even though the induced current for the short segment itself may be small (for a very short "stub"), the induced current of the long cable will still flow around the corner and exhibit equally large values (unless the shield is disconnected from ground at the sensor-end of the cable). This scenario therefore needs special considerations, in particular in the simulation code, and could be investigated alongside the future study on the coupling in presence of a parallel protective conductor (bare wire).

4. Frequent use of 0.001 and 0.0001 S/m conductivity may not align well with typical LLS locations. Starting with Fig. 7, many of the simulations employ this very-low local near-surface electrical conductivity. It would be appropriate to justify this early in the manuscript. In my experience, this value only exists for dry or non-porous rock.

Thank you for the comment. It is indeed important to clarify the choice of the ground conductivities, which where typically chosen in the range of $0.1 \cdot 10^{-3}$ - $1 \cdot 10^{-3}$ S/m. This choice is based on ITU-R Ground Conductivity Atlas (https://www.itu.int/dms_pubrec/itu-r/rec/p/R-REC-P.832-3-201202-S!!PDF-E.pdf) where most sensor locations in Austria exhibit ground conductivities within this range. Some parts of Austria exhibit higher average conductivities, exceeding $1 \cdot 10^{-3 \text{ S/m}}$ and reaching values of $10 \cdot 10^{-3}$ S/m. However, wave propagation paths across Austria often cross rocky terrain with low conductivities (see footnote on page 11 for Germany in the Atlas), and the mountainous terrain further disperses the fields. The Finnish sensor considered for the results of Fig. 19 is also located in an area with $0.3-1 \cdot 10^{-3}$

S/m conductivity. Swampy areas can exhibit ground conductivities of around σ=10·10⁻³ S/m S/m conductivity. Swampy areas can exhibit ground conductivities of around $\sigma=10\cdot10^{-3}$ S/m (respectively a resistivity of around $\rho=100$ $\Omega$m), depending on soil moisture, depth, and seasonal conditions (see, e.g., Coelho et al., 2015: Sciencedirect Link). Even higher values of about $\sigma=100\cdot10^{-3}$ S/m ($\rho = 10$ $\Omega$m) or higher have been reported in certain environments, such as Sumatra peatlands (this manuscript) measured using the Electrical Resistivity Tomography (ERT) technique (see also page 120 in this manuscript (Link) for a survey of a sludge deposit. In contrast, a survey of a garden field in Burmingham, UK (page 118 of the same manuscript) shows resistivities in the range we considered, though with pronounced inhomogeneities). Since our study focuses on various geographic areas, a range of conductivities should be taken into account in our simulation scenarios. The manuscript now addresses the choice of the ground conductivity value(s) at the end of Section 2.1 as follows: Since $\sigma_{loc}$ has a significant influence on the horizontal E-field, the coupling mechanism and, ultimately, the resulting LLS sensor site errors, values on the order of the expected (local) ground conductivities should be assumed when simulating a particular site. Although strong variations in local ground conductivities are generally expected even within small volumes near the cable (see for example (Rizki Ramdhani et al., 2020) or (Loke, 2001)), regional ranges of estimated ground electrical conductivity values are available in the World Atlas of Ground Conductivities (ITU Radiocommunication Assembly, 1999).

5. The simulations typically used a fixed propagation distance and then varied the path electrical conductivity. It might be helpful to state that for a path with fairly uniform electrical conductivity, shortening the path length by some percentage is approximately the same as decreasing the electrical conductivity in the same proportion. SO – nearby lightning will have more high-frequency content than distant lightning. This should therefore change the site error magnitude as a function of distance, given the right mix of conditions. Might this be worth stating?

Thank you for the suggestion. This was added to the document in Section 3.1 (addition highlighted in orange): As can be seen, the higher the conductivity, the lower the attenuation and dispersion. Lower values for the ground conductivity lead to more attenuated and dispersed fields with longer rise times (about 2 µs, 4 µs and 10 µs for the orange, green and red curve, respectively). It is worth noting that the frequency-dependent attenuation function (1) is also a function of distance: the farther the field propagates, the greater the attenuation and dispersion. Thus, for closer lightning strikes, the fields retain more of the high-frequency content and exhibit shorter rise times, e.g., at 50 km compared to those depicted in Fig. 7 for 100 km.

6. I am confused about Fig. 11, although I may have figured out some of the issues. In both panels in Fig. 11, the legends say "L = nnn", but I think that it should be "x = nnn". If these are really positions along the line, then it should be stated in the caption or the body of the manuscript. Also, the text states that the peak current at the line end in Fig. 11(a) is 82 mA, but it looks like 100 mA. For both Fig. 11 a &b, I am unable to reconcile the 15-20 microsecond periodicity in the current waveforms. Why does this periodicity exist for the cable grounded at both ends (11a)? This period also seems quite long, given that the round-trip time to one end and back, at the speed of light, is about 3 microseconds. A discussion of this would be helpful.

Thank you for the important input and the questions – the legend naming was indeed confusing and should read x=… which was corrected in the revised version, and the figure caption was adapted to *"Shield currents of an insulated cable of 450 m length at various locations x responding to a distant (100 km) lightning-incident field, as shown in Fig. 7 for $\sigma_p = 10\cdot10^{-3}$ S/m, $\sigma_{loc} = 10\cdot10^{-3}$ S/m. Burial depth d = 1 m."*. The peak current value of 82 mA at the line end was a remnant of a previous graph generated with a different set of parameters and was not updated to match the new graphs. Thank you for catching this! Regarding the second question, how the significant changes in the current waveform and the periodicities, it is important to highlight that the propagation speed is not the speed of light but rather the velocity of an EM field propagating in the ground with a relative permittivity of $\varepsilon_{rg}=10$, i.e.,

approximately $c_0/\sqrt{10}$ which is about one third of the speed of light. This velocity is further reduced by the dielectric properties of the insulation (cable jacket), with higher relative permittivity causing greater reduction. As reported by Bridges, 1992, the wave vector k, and thus the velocity, is further perturbed by losses, i.e., with changing electric ground conductivity $\sigma_g$. When interpreting the propagating fields, two factors must be considered: (1) This effective propagation velocity, and (2), depending on the line length, the frequency content of the impinging field (sharp versus smooth transient fields). Under certain conditions, the superposition of forward- and backward-travelling wave components may become difficult to distinguish in the resulting waveform, in particular when attenuation due to losses cause significant dispersion.

[Figure]

*Figure 1: $Z_1$ = 0 Ω (short circuit) and $Z_2$ = 1e6 Ω (open end), $\sigma_g$ = $10^{-5}$ S/m. d = 10 cm, $\varepsilon_{rd}$ = 3, $\varepsilon_{rg}$ = 10.*

We can demonstrate this in this document by presenting a plot for a very sharp transient (with sub-microsecond risetime) illuminating a cable that is located close to the surface (10 cm below ground), in a very low conductivity soil ($\sigma_g$ = $10^{-5}$ S/m). In this case, current dispersion during propagation is minimal (see Figure 1). Further, to highlight reflection phenomena, the line length is chosen to be L = 1000 m, terminated by $Z_1$ = 0 Ω (short circuit) and $Z_2$ = 1e6 Ω (open circuit). The current profile is sampled every 100 m. Under these conditions, the effects of the two traveling waves (forward and backward) and their reflections can be clearly distinguished in the current waveforms at different distances. For the simulation results shown in Fig. 11, such distinction is no longer possible due to the longer risetime, the chosen ground and line parameters, and the resulting dispersion/attenuation. Nevertheless, some reflection effects of the main travelling wave(s) are still visible in Fig. 11, for example, at about 10 µs at x=450 m, where the current drops from 90 mA to 40 mA.

7. It seems that there is a difference between the insulted wire current waveforms for L=100 in Fig. 12 (red waveform) and Fig. 14a (blue waveform). The amplitudes, fall time, and subtitles in the shape all differ.

Yes, the difference lies in the chosen set of parameters. Instead of $\rho_p$ = 10·10$^{-3}$ S/m as in Fig. 12, a value of $\rho_p$ = 1·10$^{-3}$ S/m was used. For $\rho_p$ = 10·10$^{-3}$ S/m, the waveform would have approximately matched with the L=100m case of Fig. 12, though not fully, since in Fig. 14, the incident angle was $\varphi$ = 30° relative to the cable (aligned in x-direction), instead 0°. More information on the parameters was added to the figure caption.

8. Figures 17 &18 are very nice illustrations, but the parameter domains may not be ideal. The "short" risetime associated with 0.001 S/m does not provide the likely subsequent stroke risetimes seen at 100-200 km, and the "long" risetime case seems unrealistic. Also, inclusion of 0.00001 S/m in the domain for the local conductivity is probably unnecessary, and produces unstable behaviors.

All the fields were computed assuming typical subsequent RS waveform shown in Fig. 4, which is frequently employed in the literature. The "short rise time case" of Fig. 17 associated with 0.001 S/m (i.e., about 4 µs) equals approximately the *median* rise time that the sensors measure in Austria for subsequent RSs. In general, we see little difference between subsequent RSs (median 5.3 µs) and first RSs (median 6.2 µs). The "case-study" sensor in Finland, whose site error was reproduced, even exhibits a median rise time of 8-9 µs, as stated in the caption of Fig. 19. Thus, the two risetime cases investigated in Fig. 17 and 18 align well with empirical data. The inclusion of 10$^{-5}$ S/m covers environments where rocky terrain dominates the sensor's, a scenario once observed in Mallorca, where a mountaintop sensor exhibited +/- 40° site errors. In that case, numerous metallic structures surrounded the sensor. Further, the authors do not think that instability is an issue. The purpose of Fig. 9 was to show that the computational codes that evaluate the ground penetration yield correct results (validation by means of FDTD simulations). The coupling code itself is also stable and was tested against the results from Tesche et al.'s book: EMC Analysis Methods and Computational Models, Chapter 8.6. It faithfully reproduces the currents for the case of 10$^{-5}$ S/m and a 100 m long line, see Figure 2.

[Figure]

(b) Transient Response

**FIGURE 8.20**  Frequency-domain spectrum (a) and transient response (b) for the field-induced current at $x = \mathscr{L}$ of a buried cable, for different values of earth conductivity. Double exponential excitation as defined in section 7.2.6.2.

*Figure 2: Code validation for the computation of field-to-transmission line coupling. Top graph: reproduces the results of the bottom graph (Tesche for $10^{-5}$ S/m*

**Minor/Editorial Comments/Suggestions**

9. Line 29: suggest changing "are network of sensors" to "include a network of sensors", since an LLS is more than just sensors. Changed
10. Line 33: misspelling of "Cooray". Corrected. (Thank you – this misspelling was owed to a bad entry in the literature software, which was corrected hereby as well)
11. Line 35: (picky point) - I note that Vaisala's IMPACT LLS's use the absolute time of arrival, relative to the estimated discharge time, rather than time-differences between sensors. Thank you for the comment – this information will however not be included in the paper.
12. Line 48: suggest adding "additive" after "spurious". Great idea! This was added.

13. Lines 49-50: suggest eliminating "at the sampling instants", since the spurious fields superimpose on the whole waveform. Removed as suggested
14. Line 54: suggest adding "nearby" before "ground". Added
15. Line 69: suggest changing "serves as an estimator for" to "is used to produce the estimated" Adapted
16. Line 72: The second half of this line, starting with "respectively", does not seem to fit here. Wrote "as the results presented in Section 3.3 of the present study show" and removed it from the parenthesis
17. Line 105: suggest changing "shield currents" to "cable currents" or something like this, since a bare wire is one of the conditions. Adapted
18. Line 108: suggest changing "MDF techniques to locate lightning." to "magnetic fields to locate lightning and/or estimate peak current." Adapted
19. Line 112: suggest adding "cable grounding method" to the list Thank you for the suggestion! Was added to the list
20. Line 249: suggest changing "introduces an error to" to "adds a spurious term beyond", or something like this, since it does not change the true incident field. Adapted
21. Line 326: unnecessary line break after "shows". Yes, this was owed to a graphic previously placed there. Thanks, the line break was removed.
22. Line 329: should this be "line end", and not "cable end" ? Better indeed! Was adapted
23. Line 330: Should the percentage be 60%, rather than 75%? Thank you for pointing out this error. It should have indeed read 60%!
24. Lines 336-7: suggest changing "local conductivity on the site reduces" to "local conductivity is lower". Yes, this is better. Was adapted. Thank you!
25. Line 351: suggest changing "distance of the MDF to" to "vertical separation between the MDF antennae and". Adapted
26. Fig. 15: I cannot find the cable length that was used for this study Thank you for the note. It was L = 100 m. The info is now contained in the figure caption
27. Caption for Fig. 13: The first line of the caption says "site errors", but it is actually the magnetic fields Correct. Magnetic field is more accurate!
28. Line 364: might want to say "contributes to H_sampled" rather than H_y, since the scattered field does not change H_y. Indeed, the wrong word was chosen. Correct would have been "is a y-component that adds to" instead of "contributes to". Corrected!
29. Lines 377 to 381: This might be a good place to note that there will also be high frequency content in the incident field when the propagation distance is short, as long as the sensor bandwidth accommodates the higher frequencies. Thank you for the suggestion, a sentence was added as follows (addition highlighted in orange): While incident fields with very high-frequency content (i.e., short rise times), combined with very low local ground conductivity $\sigma_{loc}$ *and* long cables, may occur in reality, such scenarios are rare. Nevertheless, this possibility should not be overlooked, because, as explained in Section 3.1, lightning discharges occurring close to the sensor also contain high frequency content, and thus short measured rise times can be expected.
30. Line 390: suggest changing "such as" to "including", since this is what you have done. Adapted
31. Line 416: The text states "distance to the cable", but it would be more precise to say "vertical distance to the cable." This wording occurs in other places in the manuscript Adapted throughout the work

32. Line 434: It would be clearer to say "propagation path ground conductivity" instead of "ground parameters" Partly adapted (only, if just $\sigma_{loc}$ was concerned, but $\varepsilon_{r,g}$ not)

33. Line 443: does this simply repeat what is stated at the end of the previous paragraph? Good observation. There was indeed some repetition present. The sentences/paragraphs were merged and part of it removed. Having both ends grounded, where the largest currents can flow, represents the worst-case scenario for LLS sensor site errors. This analysis deliberately focuses on this worst-case scenario with the aim of understanding the primary interrelation between the influencing parameters.

34. 467-469: Does this statement also apply to the bare conductor case? This is explained in the last bullet point

35. Line 486: suggest adding "local" before "ground" (also in other places in the manuscript) Adapted

36. Line 487: suggest changing "left to the peak" to "left of the peak" or "before the peak" Adapted

37. Line 545: should probably say "above the cable", rather than "from the cable". This addresses my Comment 1 above! Might not be needed if you know the answer to my Comment 2. The sentence was slightly reformulated: Future studies should consider the impact of a follow-on bare wire, such as horizontal electrode placed in close distance above or next to the cable. In (Theethayi and Thottappillil, 2007), a follow-on bare wire in a horizontal distance of 10 cm was shown to significantly reduce the internal voltages between the core and the cable shield.

38. Line 556: please add "local" before "conductivities" Adapted

39. Lines 562-3: should this be clear about the difference between grounding at both ends, vs. just grounding at the end that is far from the sensor? You discuss this at Line 613-615, but it is not made clear as part of the study. "the difference between a grounded and a floating cable shield at the sensor end," has been added to the list. Thank you for the note!

---

## Author Comment (AC4)

**General Comments**

This study provides an interesting and thorough analysis on the physical mechanisms behind angle and amplitude site errors for magnetic direction finders in lightning location systems. This work is important for understanding the potential errors due to sensor and cable placement associated with sensor site locations. The authors provide a clear and concise description of their methodology towards evaluating sensor site errors and determining which variables are most important to their analysis. I don't have any major issues with this work but I have a few minor comments to add without repeating what the other reviewers have suggested.

The authors want to thank Dylan Goldberg for the valuable time to review the manuscript and appreciate the efforts to address some issues.

- Line 114: Is the theoretical shielded conductor single or double shielded? Do the results of this study apply to both? It would also be interesting for a future study to see if the results vary between types of shielding used (i.e. braid vs foil). In fact, a single insulated conductor is considered as a proxy to a shielded cable. The difference in impedance between a solid conductor and a conductor shield (e.g., braided shield) was investigated and found to be negligible, having no significant impact on the results. A future study will explore various cable types, including double shielded cables.

- Lines 204 - 205: Should "finite-length" be removed since that (at least to me) is implied by a cable of length L? Thank you, this was corrected.

- Lines 244 - 245: It's stated that contributions beyond 50 m are assumed to be negligible. Was this number decided based on a percentage from the 1/r^2 dependency or were contributions beyond this number tested initially? This was tested against the sum of all current contributions, i.e., the spurious magnetic field $H_{err}$. Beyond 50 m, contributions of the cable/conductor current altered $H_{err}$ by less than 1%.

- Figure 7: The ground electric conductivity for the red line should also have units in the legend to be consistent with the other lines. Thank you for the observation. This issue was already addressed in response to the first reviewer's comments, and the correction has been incorporated into the revised manuscript.

- Figure 8: The vertical axis labels are missing the closing parenthesis around the units. Also, should the d = -0.0 m burial depth be negative? Thank you very much, that had gone unnoticed! The figure has been corrected.

- Figure 9: It may be beneficial to use the same notation for the conductivities across the different figures and text; maybe switch the legend conductivities from scientific notation to magnitudes as shown in the figure caption? Also, I'm not sure I understand the change in the time scale of the horizontal axis for this figure compared to figures 7 and 8. The change in the time scale was due to a slightly different set of scripts (from earlier work) that were employed to do the validation. As a result, it remained unnoticed that the x-axis still showed the absolute time of arrival of the field. Since this was unfortunately not a self-contained script but a manual edit, the following note was added in the caption to clarify this: *The time-axis represents the absolute time of arrival of the EM field at a distance of 100 km (approximately 333 μs).*

- Line 308: The equation "d=1m" should probably be "d = 1 m" (spaces added) to be consistent with the other inline equations. This also occurs in the legends of some figures. Thank you for the note. This was corrected.

- Line 326 - 327: There seems to be an extra line/paragraph starting here. Thank you for the observation. This was owed to a formatting issue with MS Word. It is corrected in the revised version.

- Figure 19: Fig (a) appears to be lower in resolution (dpi) quality than the other figures in this paper. True. Unfortunately, this is one of the graphics that are difficult to access for direct modification. For aesthetic improvement, the axes were manually redrawn.